

# New insights into the decadal variability in glacier volume of an iconic tropical ice-cap explained by the morpho-climatic context, Antisana, (0°29' S, 78°09' W)

Rubén Basantes-Serrano[1,2], Antoine Rabatel[3], Bernard Francou[3], Christian Vincent[3], Alvaro Soruco[4], Thomas Condom[3], Jean Carlo Ruíz[3,5]

[1]Facultad de Ciencias de La Tierra y Agua, Universidad Regional Amazónica Ikiam, Tena, Ecuador
[2]CAOS, Observatorio del Clima, Quito, Ecuador
[3]Univ. Grenoble Alpes, CNRS, IRD, Grenoble INP, Institut des Géosciences de l'Environnement (IGE, UMR 5001), F-38000 Grenoble, France
[4]Instituto de Investigaciones Geológicas y del Medio Ambiente, Universidad Mayor de San Andrés, La Paz, Bolivia
[5]Sorbonne Université, UMR 7619 METIS, Case 105, 4 place Jussieu, F-75005 Paris, France

*Correspondence to*: Rubén Basantes-Serrano (ruben.basantes@ikiam.edu.ec)

**Abstract.** We present a comprehensive study of the evolution of the glaciers in the Antisana ice cap (tropical Andes) over the period 1956-2016. Based on geodetic observations of aerial photographs and high-resolution satellite images, we explore the effects of morpho-topographic and climate variables on glacier volumes. Contrasting behaviour was observed over the whole period, with two periods of strong mass loss: 1956-1964 and 1979-1997, and two periods with slight mass loss: 1965-1978
and 1997-2016. There was a 42% reduction in the total surface area of the ice cap. Individually, glacier responses were modulated by morpho-topographic variables (*e.g.,* maximum and median altitude, and surface area), particularly in the case of the small tongues located at low elevations (Glacier 1, 5 and 16) which have been undergoing accelerated disintegration since the 1990s, and will likely disappear in the coming years. Moreover, thanks to the availability of aerial data, a surging event was detected in the Antisana G8 in the 2009-2011 period; such an event is extremely rare in this region and deserves a dedicated
study. Despite the effect of the complex topography, glaciers react in agreement with changes in climate forcing, with a stepwise transition towards warmer and alternating wet/dry conditions since the mid-1970s. Long term decadal variability is consistent with the warm/cold conditions observed in the Pacific Ocean represented by the Southern Oscillation Index.

## 1   Introduction

The tropical Andes have experienced continuous atmospheric warming over the last four decades (Chimborazo et al., 2022).
Although the pattern is not identical from one site to another (Aguilar-Lome et al., 2019; Pepin et al., 2015; Yarleque et al., 2018) it confirms the marked glacier shrinkage observed in this part of the Andes (Dussaillant et al., 2019; Masiokas et al., 2020). Furthermore, climate projections warn that warming will continue during the 21st century (Bradley et al., 2006; Urrutia



and Vuille, 2009), thereby increasing the rate of glacier melt, leading to major shrinkage and even to the complete disappearance of many glaciers in the region in the coming decades (Rabatel et al., 2018; Yarleque et al., 2018).

Despite the atmospheric warming, the small  Antisana 15α Glacier (0.3 km² in 2016) in Ecuador (Figure 1), which has been surveyed since the mid-1990s, showed only a slight mass loss during the 1995-2012 period (Basantes-Serrano et al., 2016). Antisana 15α is part of an ice cap and thanks to its very high elevation (up to 5,700 m a.s.l.), the glacier still has a large accumulation area (roughly 60% of its total surface area), which is not the case of the vast majority of the small glaciers in this region (Rabatel et al., 2013b). These peculiar morpho-topographic characteristics associated with the almost year-long humid

conditions in the inner tropics (which contribute a considerable amount of solid precipitation in the upper reaches of the glacier) are the main drivers of the limited mass loss of the Antisana 15α Glacier (Basantes-Serrano et al., 2016). However, despite being the longest glaciological time series in the inner tropics, the representativeness of the changes observed on Antisana 15α Glacier needs to be assessed at the scale of the other small glaciers. To this end, high resolution stereoscopic imagery is useful to evaluate the representativeness of the glacier mass changes at the scale of a mountain range (e.g., Rabatel et al., 2006;

Soruco et al., 2009b).

In addition, field measurements taken on Antisana G15 glacier showed that the key periods of the year that explain more than 90% of the variance of the mass balance are centred on the equinoxes, i.e.  between March and May (MAM) and between September to November (SON) (Basantes-Serrano et al., 2016; Francou et al., 2004). Indeed, these quarters are those with maximum potential solar radiation and higher rates of precipitation caused by the transit of the ITCZ (Vuille et al., 2000).

Previous studies also confirmed a strong relationship between the variability of the mass balance and the occurrence and intensity of ENSO events at an annual time step (Favier et al., 2004; Francou et al., 2004; Rabatel et al., 2013b). Consequently, we expected a cumulative influence of seasonal climate conditions in the MAM and SON quarters on the long-term variability of the decadal mass balance.

The present study is the first comprehensive assessment of the multi-decadal variability of the glacier mass fluctuations

quantified on 17 glaciers covering an equatorial ice cap using the geodetic method. We explore the role of topographic features in determining the responses of individual glaciers and the impact that the change in surface area caused by the geometric adjustment of small glaciers, may have on the mass balance of the ice cap as a whole. We then discuss the changes in ice masses resulting from the climatic conditions that have prevailed in the region since the mid-1950s. The paper is structured as follows: Section 2 describes the study site and the key climate features of the inner tropics. Section 3 provides a concise

summary of remote sensing and climate data and data processing. Section 4 provides a detailed description of the changes observed in the glaciers in five sub-periods since the mid-1950s and the topographic-climatic drivers that may explain the glaciers' responses together with the potential influence of the large-scale ENSO episodes on the decadal variability of the tropical ice cap.

Our study provides a baseline to model the future effects of glacier retreat on the water cycle and mountain ecosystems in the

inner tropics.


## 2    Study site and climate settings

**Figure 1 (a)** The Antisana ice-cap with its 17 glaciers. The blue polygon located Glacier 15 where in-situ observations have been available since 1995. The grey line represents the glacier boundaries in 1956 and the black lines represent the boundaries in 2016. Data are missing on the southeast side of the ice cap due to the cloud cover (area shown in white). False-colour (RGB-324) pan-sharpened composite of Pleiades 0.5 m ortho-images used as the background show the distribution of the vegetation in the surrounding glaciers (© CNES 2016, Distribution Airbus D&S). The bar charts in **(b)** and **(c)** show precipitation seasonality throughout the year in the 1950-2018 period measured at two weather stations, M003 (resp. M188) located on the western (resp. northern) side of the volcano (see Fig. 2). Panels **(d)** and **(e)** are the seasonal mean precipitation maps for December–February (DJF) and June–August (JJA) derived from ERA5 reanalysis data for the 1950-2018 period (shading scale). The solid black line shows the average position of the ITZC estimated from the ERA5 precipitation rates and the South American Monsoon System (SAMS, blue arrows) during the austral summer (DJF). The red star locates Antisana volcano.





The Antisana is an active stratovolcano capped by a ~13 km² ice-cap in 2016, located close to the equator (0°29'S; ~5,700 m a.s.l) around 40 km to the east of the city of Quito. Little information is available concerning the volcano's activity in the last

400 years, no recent morphological changes in the Earth's crust, thermal activity or local ice reductions due to hot streams have been identified that suggest recent volcanic activity (Hall et al., 2017). The ice cap has an overall radial structure composed of 17 glacierised catchments identified through photogrammetric restitutions following the boundary of the water basins from the summit of the volcano to the valleys. Some of the glaciers have a local name but most are named by numbers starting from the north and going clockwise. Hereafter, we use the numerical nomenclature for easy identification of the glaciers, for example,

Glacier Antisana 15 is noted G15 (Figure 1a).

The climate features in the tropical Andes are mainly driven by the oscillatory position of the Intertropical Convergence Zone (ITCZ) and the presence of the South American Monsoon System (SAMS) illustrated in Figure 1d and Figure 1e, and the deep atmospheric convection originating from the very wet Amazon rainforest which create precipitation regimes in this region which differ even at short distances (Figure 1b, c). The Andes Cordillera controls the spatial patterns of the wet flows driven

by the trade winds blowing from the Amazon basin (Espinoza et al., 2020; Garreaud, 2009). In fact, the site is characterised by a mixed seasonal rainfall cycle. On the western side of the volcano, a bimodal cycle is observed with a first maximum from March to May and a second maximum in November (Figure 1b), whereas the eastern side shows unimodal seasonality with a maximum lasting from July to August (Figure 1c). The climate patterns may occur simultaneously when the Amazon seasonality penetrates the Andes cordillera rather deeply through the wide valleys due to the atmospheric circulation (Tobar

and Wyseure, 2018).

Local precipitation occurs all year round and ranges from ~1,300 mm yr⁻¹ on the western slope to ~3,000 mm yr⁻¹ on the eastern slope of the volcano (Figure 1b, c), whereas air temperatures display more regional behaviour and remain almost constant throughout the year. Furthermore, *in-situ* meteorological observations recorded at elevations ranging from 1,900 to 5,000 m a.s.l. suggest a positive vertical precipitation gradient, with annual precipitation rates that can reach a maximum of ~6,000

mm yr⁻¹ (Emck, 2007; Garreaud, 2009) and thus have a significant effect on the accumulation rates over the course of the year (Basantes-Serrano et al., 2016). An increase (decrease) in air temperature causes rainfall (snowfall) thereby enhancing ablation (accumulation) processes at the surface of the glacier (Favier et al., 2004; Francou et al., 2004). In addition, the response of the glacier to climate variation is modified during ENSO events, with very positive (resp. negative) mass balances during La Niña (resp. El Niño) events (Francou et al., 2004).

Morpho-topographic-climate interactions play an important role in the behaviour of glaciers but also in the definition of the biogeography of the site (Figure 1). Below 4,750 m a.s.l., glacier retreats have left behind several morainic deposits on the western slope of the volcano, which become new ecosystems for certain plants and other specialised organisms capable of developing in particularly challenging environmental conditions (Cauvy-Fraunié and Dangles, 2019; Cuesta et al., 2019). Below 4,650 m a.s.l., small *páramo* vegetation dominates the landscape. On the eastern side, glaciers extend down to 4,400 m

a.s.l., and the areas abandoned by the glacier are immediately occupied by a dense rainforest and shrub vegetation.



## 3  Data and Methods

### 3.1  Processing of aerial and satellite data

Photogrammetry is a powerful tool to assess the glacier-wide mass balance of several glaciers at the scale of a mountain range.

In the present study, we took advantage of the availability of high-resolution aerial and satellite stereoscopic data from different sources acquired at different dates. The stereoscopic data enabled us to map every glacierised basin on the Antisana ice cap at high spatial resolution (Table 1), and to quantify changes in the mass balance of the glaciers at a decadal time step since 1956.

**Table 1** High-resolution remote sensing data used in this study. (*) geodetic reference used to evaluate the xyz-consistency of the elevation data set.

| Date | Sensor | Focal (mm) | Ground pixel size (m) | Geometric adjustment ($\sigma x$, $\sigma y$, $\sigma z$) (m) |
|---|---|---|---|---|
| February 15, 1956 | T-11 | 153.55 | 0.64 | 0.36 , 0.22 , 0.40 |
| February 7, 1965 | KC1B | 151.68 | 0.46 | 0.29 , 0.29 , 0.37 |
| January 18, 1979 | RC10 | 152.68 | 0.74 | 0.30 , 0.18 , 0.33 |
| August 3, 1997 | RC30 | 152.91 | 0.39 | 0.21 , 0.27 , 0.20 |
| September 13, 2009* | RC30 | 152.89 | 0.48 | 0.20 , 0.29 , 0.12 |
| December 12, 2016 | Pléiades | - | 0.50 | - |


On the one hand, we used 90 raw aerial photographs for the Antisana volcano taken at five different dates in 1956, 1965, 1979, 1997 and 2009. The aerial surveys were conducted by the *Instituto Geografico Militar* (IGM) of Ecuador. The bundle block adjustment was conducted based on a geodetic network of 21 ground control points (GCPs) collected using the DGPS method in a static mode with an uncertainty of less than ~0.20 m (see Figure 1b in Basantes-Serrano et al., 2016). For each aerial

survey, very dense 3-D point clouds representing between 90,000 and 185,000 points, were generated automatically on the non-glacierised areas by applying an image correlation technique which matches similar features in the overlap area between two images (Hirschmuller, 2005). Over the glaciers, where the snow cover, shadows and cloud cover prevented effective application of the correlation methods, 42,500 3D-points per aerial survey were measured randomly by photogrammetric restitution, giving a total of 212,500 points, avoiding areas with low contrast and saturated areas. These 3-D point clouds were

subsequently used to compute differences in elevation in each study period. All the photogrammetric tasks were performed IMAGINE *Photogrammetry* (formerly the Leica Photogrammetry Suite (LPS) (Hexagon Geospatial®).

On the other hand, a pair of Pleiades images of the same area taken from different perspectives enabled the geodetic mass balance survey to be extended to 2016. These images were acquired by the sun-synchronous Pleiades-1A platform with a viewing angle of 12.69° and 8.76° in the across track direction ($\omega$), and of -8.66° and 10.16° in the along track direction ($\phi$)

with respect to the nadir. To check the ability of the stereoscopic pair to capture the mountain topography, we computed the ratio of the distance of two successive positions of the satellite ($B$) to its altitude above ground $H$, resulting in 0.33, which is optimum for processing numerical models in mountainous areas (Perko et al., 2019). The relative orientation of the Pleiades images was performed using the rational polynomial coefficients (RPCs) provided by the ancillary information. Finally, stereo-



matching algorithms enabled the generation of a 3D point cloud by means of pixel-to-pixel correspondences between the
images. The 3D mapping procedure for the stereoscopic pair was performed using the Ames Stereo Pipeline (ASP), a free
open-source software developed by the National Administration Space Agency (Beyer et al., 2018). The Pleiades images were
acquired in the framework of the ISIS programme of the French Space Agency (CNES) in the framework of the Pleiades
Glacier Observatory initiative (Berthier et al., 2014).

### 3.2  Geometric adjustment of the geodetic data

To assess spatial consistencies in the geodetic information, surface elevations in stable rocky areas in the forelands of the ice-
cap were compared following the iterative workflow proposed by Nuth and Kääb, (2011). Co-registration is recommended
before identifying changes in glacier elevation to ensure precise $xyz$ alignment between the geodetic data. A conical structure
like the Antisana volcano is perfect to correct elevation bias because, as long as there are sufficient examples of differences in
elevation (hereafter $dh$-samples) over stable rocky areas, it covers all terrain aspects. The elevation differences ($dh$) of the two
misaligned DEMs are a function of the slope of the terrain ($\alpha$) and the aspect ($\omega$) of the $dh$-samples as follows:

$$\frac{dh}{\tan(\alpha)} = a \cdot \cos(b - \omega) + c \qquad (1),$$

where $a$ is the magnitude of the misalignment, $b$ is the direction of the misalignment, and $c$ is the mean elevation bias. Then,
the $\Delta xyz$ adjustment factors are computed as follows:

$$\Delta_x = a \cdot \sin(b) \qquad (2),$$
$$\Delta_y = a \cdot \cos(b) \qquad (3),$$
$$\Delta_z = \frac{c}{\tan(\bar{\alpha})} \qquad (4),$$

As the geometric characteristics of the 2009 geodetic survey were optimal, the digital elevation model (DEM) generated from
these data was used as a reference to evaluate the bias in the geodetic data of the other acquisitions. Because high spatial
resolution data were used in this study (Table 1), an elevation difference layer of 10x10 m cell size was computed for each
geodetic comparison using the universal kriging technique based on an exponential semi-variogram model. After co-
registration, analysis revealed that the geodetic datasets were slightly unaligned with respect to the reference, with a mean
elevation difference ranging from -1.8 to 3.2 m on stable terrain. Consequently, we decided to perform 3D geometric
adjustment to minimise the difference in the geodetic data. The results are presented in Table 2 and in supplementary materials.
In all cases, there was a reduction of around 8% in the standard deviation and the mean difference in elevation was improved,
and a normal distribution of the $dh$ on the glacier forelands is better shaped with an average standard error of 0.03 m after 3D
geometric adjustment (Fig. S1 in supplementary materials).


**Table 2** Differences in average elevation and its standard deviation over glacier free terrain before [$\overline{dh}'$; $\sigma'$] and after [$\overline{dh}''$; $\sigma''$] the 3D geometric adjustment, computed for the total number of elevation differences ($dh$-samples) off glacier for the five study periods. The geometric solution in the three axes [$\Delta_x, \Delta_y, \Delta_z$] and the average standard error after adjustment [Se] are given. All the statistics are expressed in metres [m].

| Period | $\overline{dh}'$ [m] | $\sigma'$ [m] | $\Delta_x$ [m] | $\Delta_y$ [m] | $\Delta_z$ [m] | $\overline{dh}''$ [m] | $\sigma''$ [m] | Se [m] | $dh$-samples off glacier |
|---|---|---|---|---|---|---|---|---|---|
| 1956-2009 | -1.35 | 12.04 | -1.25 | 7.22 | -1.03 | 0.29 | 11.95 | 0.04 | 94,851 |
| 1965-2009 | -1.33 | 15.89 | -6.36 | 6.89 | 0.51 | -0.71 | 15.05 | 0.05 | 102,229 |
| 1979-2009 | 2.69 | 5.57 | 0.77 | 6.26 | 1.53 | 0.64 | 5.00 | 0.02 | 95,721 |
| 1997-2009 | 3.18 | 5.45 | 1.71 | 6.64 | 0.82 | 0.65 | 5.02 | 0.01 | 146,870 |
| 2009-2016 | -1.79 | 4.06 | 7.50 | 0.79 | 0.40 | -0.47 | 4.02 | 0.01 | 184,881 |

The co-registration results showed a similar displacement vector of around 8 m in all five datasets. For the aerial-based geodetic data, this was attributed to the number and distribution of the GCPs used to solve the stereo orientation. For the Pleiades-based data, this is probably explained by the fact that the triangulation and geometric adjustment of Pleiades stereo-images was based on the RPC model without including GCPs. The whole elevation data can thus float up to 10 m above or below the ground and has to be adjusted horizontally and vertically (Berthier et al., 2014).

### 3.3 Computing the geodetic mass balance

The geodetic method captures multidecadal changes in glacier mass (Cogley et al., 2011), and also makes it possible to document the spatial variability of changes in elevation at the surface of the glaciers. On the basis of this variability, we spatially optimised the 3D measurements at the surface of glaciers to compute changes in elevation using the geo-statistical framework proposed by Basantes-Serrano et al. (2018). The number of $dh$-samples per glacier depends on their size, a minimum of 1,000 sampling points per km$^2$ was thus measured to estimate the glacier mass balance. With this approach, instead of trying to cover the entire glacier surface with 3D-coordinates, we chose sites where significant elevation changes are expected, avoiding places with steeper slopes where large elevation outliers are probable (Berthier et al., 2004).

No measurements were made on about 10% of the glacier surface comprising areas with low contrast, including saturated areas, or with a slope greater than 45°. $dh$ outliers were removed considering a threshold of three normalized median absolute deviations (NMADs), as done in other studies (e.g., Braun et al., 2019; Brun et al., 2017). The procedure for removing outliers was applied to each individual glacier, and about 5% of the differences in elevation outside the 3NMAD threshold were excluded. Based on the remaining $dh$-samples, an exponential variogram model was fitted to generate an elevation difference grid of 10x10 m cell size using the universal kriging technique (Basantes-Serrano et al., 2018). Because all gridded surfaces have the same spatial resolution, we did not expect any curvature error in the topography of glaciers that could affect the estimation of the geodetic mass balance (Gardelle et al., 2012). Finally, the glacier-wide mass balance expressed in meters of water equivalent (m w.e.) for each glacier was computed by:





$$B_g = (\bar{\rho} * r^2 * \sum_{i=1}^{p} \Delta h(x_i))/\bar{S} \qquad (5),$$

where $\bar{\rho} = 850$ kg m$^{-3}$ is the average density value recommended by Huss (2013), $r$ is the pixel size, $\Delta h$ is the change in glacier

surface elevation at each location $x_i$, $p$ is the number of pixels covering the glacier at its maximum extent, and $\bar{S}$ is the glacier

surface area averaged over the period between the date of the first aerial survey and date of the second aerial survey. The mean

annual rate of glacier-wide mass balance $\dot{B}_{g.t}$ considering the number of years $N$ for each study period is then calculated as:

$$\dot{B}_{g.t} = \frac{B_g}{N} \qquad (6).$$

## 3.4 Uncertainty analysis

To determine uncertainty in the geodetic mass balance, we assessed the contribution of the following individual sources:

- First, the uncertainty related to the DEMs used to compute the difference in elevations following the method proposed by
  Rolstad et al., (2009). Thus, we estimated the spatial auto-correlation of the grid of $dh$ based on a semi-variogram analysis.
  For the Antisana ice cap, we found that the difference in elevation in non-glacierised areas is spatially uncorrelated beyond

a horizontal distance ($a_1$) of about 1,250 m in all geodetic surveys; this distance was computed from the variogram model.
  Because the averaging of the glacier areas in the first survey $S_T$ is greater than the effective correlated area $s_{cor} = \pi * a_1^2$
  in all five periods, the uncertainty of spatially averaged differences in elevation in metres is defined by:

$$\sigma_{dh.g} = \pm\sigma_{dh} * \sqrt{\frac{S_{cor}}{5*S_T}} \qquad (7),$$

where $\sigma_{dh}$ is the standard deviation of $dh$ over stable terrain in the vicinity of the ice cap.

- Second, regarding the uncertainty of the average density value, we used the one recommended by Huss (2013): $\sigma_\rho = \pm 60$
  kg m$^{-3}$.

- Third, the internal ablation due to the heat transfer in the subglacial interface layer and due to heat released due to glacier
  dynamics needs to be taken into account. These components vary in space and time and may be significant for glaciers
  located at the summit of Andean volcanoes, where steeper slopes predominate. Because of the lack of data concerning

basal melting for Antisana glaciers, we assumed a maximum error of internal ablation $\sigma_{i,abl}$ equal to $\pm 0.04$ m w.e. yr$^{-1}$
  estimated by Basantes-Serrano et al. (2016) for glacier Antisana G15.

- Fourth, missing glacier elevation differences, i.e., where no elevation measurements were available, were attributed to: i)
  saturated areas because of the coarse grain size (low sensitivity) of the film used during the aerial surveys or the presence
  of bright snow patches, most of which were located in the lower flat areas; ii) cloud cover which affects some parts of the

glaciers; and iii) sites with slopes greater than 45°. These three characteristics are mostly found in the accumulation zone
  of the Antisana ice cap. Recently McNabb et al., (2019) explored the effect of DEM voids on the geodetic mass balance
  and evaluated several interpolation methods to fill gaps in elevation data. These authors concluded that interpolation
  methods work quite well when data gaps are limited to small patches or when elevation changes are regularly distributed



over the glaciers, as is the case in ablation zones. However, this is not necessarily true for the accumulation zones of
Antisana glaciers, which present a noisy spatial pattern of either negative or positive surface elevation changes, linked to
the interaction between ice dynamics and the harsh topography of the volcanic cone (Basantes-Serrano et al., 2016). Thus,
rather than filling $dh$ voids using an interpolation method, which can introduce biases on the geodetic mass balance, we
decided to estimate the uncertainty expressed in metres of water equivalent (m w.e.) by the expression:

$$\sigma_{Svoid} = \sigma_{dh.g} * (\overline{\rho}) * (\frac{S_{g.void}}{S_g}) \qquad (8),$$

where $S_{g.void}$ is the portion of individual glacier with missing $dh$ values, and $S_g$ is the total surface area of the
corresponding glacier recorded during the first aerial survey.

-    Finally, to match the arbitrary time scale imposed by geodetic surveys and the hydrological time scale, it is necessary to
perform a temporal homogenisation that accounts for the ablation and accumulation processes which occur during the
time lag (Thibert et al., 2008). Indeed, the dates of the aerial surveys used in this study (Table 1) do not match the
hydrological year in the inner tropics (i.e. from January to December). However, no meteorological and glaciological
observations at annual time step, are available in this region before the mid-1990s (Basantes-Serrano et al., 2016),
therefore no time adjustment or time bias estimations were possible and we kept the original dates for the mass balance
estimations.

Thus, the overall uncertainty in the geodetic mass balance of each individual glacier expressed in metres of water equivalent
(m w.e.) was estimated using the propagation error approach of uncorrelated variables as follows:

$$\sigma_{B_g} = \sqrt{\sigma_{dh.g}^2 * (\overline{\rho}) + \sigma_{\rho}^2 + \sigma_{i,abl}^2 + \sigma_{Svoid}^2} \qquad (9).$$

### 3.5  Climate variability and local drivers of glacier response

Meteorological observations are relatively recent in Ecuador and are very scarce above 4,800 m a.s.l. Climate reanalysis
provides an opportunity to get key information to reconstruct climate conditions in a specific region, and to help understand
the impacts of climate changes in the past when direct observations are scarce (Hersbach et al., 2020). To analyse the glacier
response to regional climate variability, we considered monthly interpolated air-temperature and precipitation data at different
pressure levels from the fifth generation of ERA5 reanalysis provided by the European Centre for Medium-range Weather
Forecasts (ECMWF), which is a 0.25° (~30 km) grid resolution. ERA5 reanalysis replaces the former version of ERA-Interim
reanalysis.
Reanalysis data are prone to systematic biases due to the limited spatial resolution and simplified physics used to represent
climate, particularly in mountain areas where orographic factors drive climate conditions. In practice, temperature
(precipitation) measurements made near glaciers are somewhat higher (lower) than reanalysis data. To adjust the ERA5 dataset
to observations, we compared the reanalysis data close to the grid cell to the station location and the *in-situ* meteorological
data at a monthly time step for the nine weather stations with the longest series that recorded climate variability from the





western and the eastern sides of the cordillera between 1950 and 2018 (Figure 2 and Table 3). This observational network is located below ~4,500 m a.s.l, and is maintained, homogenised and quality controlled by the *Instituto Nacional de Meteorología e Hidrología* (INAMHI).

This comparison showed that ERA5 is able to capture the western climate signal observed at M003 weather station (Izobamba), located 50 km from the volcano. The average annual rainfall in M003 is 1,446 mm yr$^{-1}$ (±15%), not far from the annual average

recorded by one of the rain gauges installed on the western slope of Antisana, close to the terminus of the glacier G15, i.e., an average of 1,235 mm yr$^{-1}$ between 1995 and 2018. On the eastern slope, ERA5 reanalysis was unable to capture the seasonality of precipitation of any of the weather stations located in the Amazon basin. However, it will be recalled that the estimated precipitation on the Amazonian side can reach 3,000 mm yr$^{-1}$. On the other hand, the temperature recorded by the M188 weather station (Papallacta) was satisfactorily captured by ERA5 reanalysis.

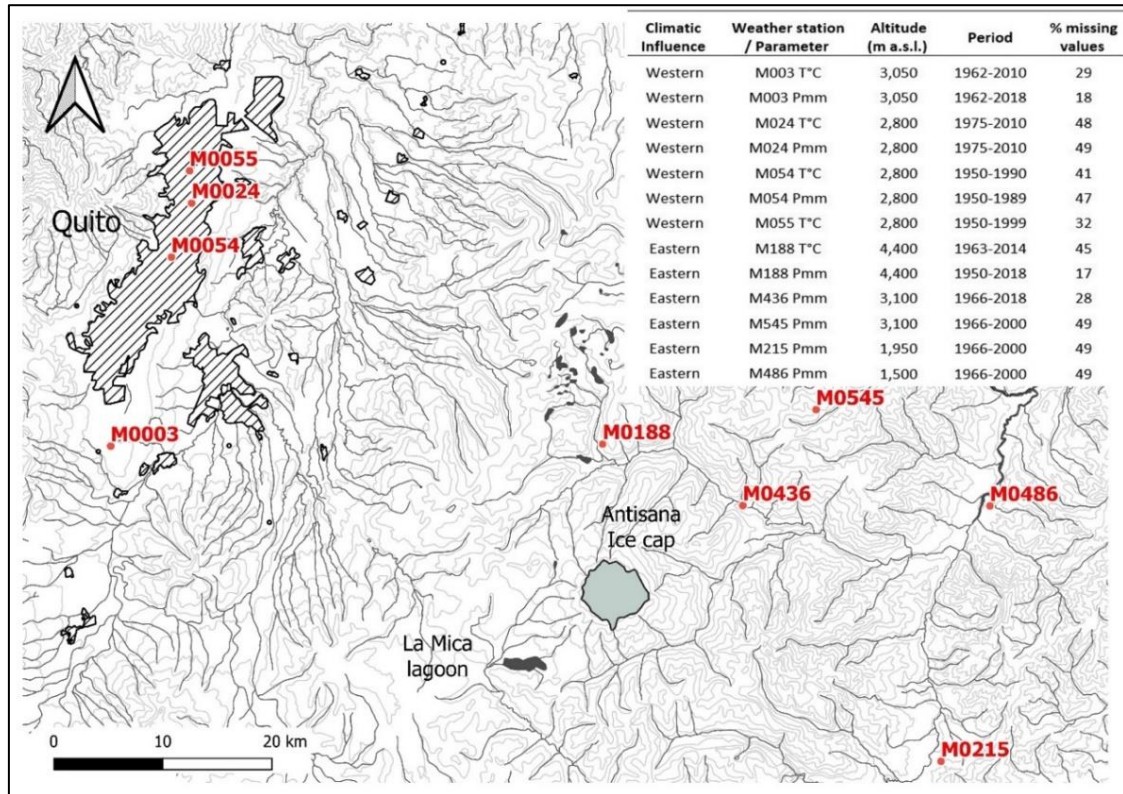

| Climatic Influence | Weather station / Parameter | Altitude (m a.s.l.) | Period | % missing values |
|---|---|---|---|---|
| Western | M003 T°C | 3,050 | 1962-2010 | 29 |
| Western | M003 Pmm | 3,050 | 1962-2018 | 18 |
| Western | M024 T°C | 2,800 | 1975-2010 | 48 |
| Western | M024 Pmm | 2,800 | 1975-2010 | 49 |
| Western | M054 T°C | 2,800 | 1950-1990 | 41 |
| Western | M054 Pmm | 2,800 | 1950-1989 | 47 |
| Western | M055 T°C | 2,800 | 1950-1999 | 32 |
| Eastern | M188 T°C | 4,400 | 1963-2014 | 45 |
| Eastern | M188 Pmm | 4,400 | 1950-2018 | 17 |
| Eastern | M436 Pmm | 3,100 | 1966-2018 | 28 |
| Eastern | M545 Pmm | 3,100 | 1966-2000 | 49 |
| Eastern | M215 Pmm | 1,950 | 1966-2000 | 49 |
| Eastern | M486 Pmm | 1,500 | 1966-2000 | 49 |


**Figure 2** Map of long-term weather stations located in the region. The city of Quito and the ice cap are represented by the shaded and solid grey polygons, respectively. The inset table lists the meteorological stations and the variables used in this study, the percentage of missing values at a monthly time step are reported. Cartographic information was provided by the Instituto Geográfico Militar of Ecuador (IGM).

To reduce the bias between the ERA5 data and reference observations, we retain M003 data and we applied the following

approaches: (i) for temperature calibration we applied a Bias-Correction method which uses the mean differences and variability of the data series (Hawkins et al., 2013); and (ii) for precipitation calibration, we applied a Quantile Mapping approach which uses a statistical transformation function to adjust the distribution of ERA5 data to match the distribution of



observations (Thrasher et al., 2012). To define the function, we applied a non-parametric transformation termed QUANT in the R package QMAP (Gudmundsson et al., 2012). Once biases were reduced, a simple statistical metric enabled us to confirm
that the mean difference between ERA5 data and *in-situ* observations was improved by 90% and the RMSE reduced by 60%, showing that the calibration methods performed quite well except for the RMSE of M188 station, which showed no improvement (Table 3).

Table 3 Temperature (T°C) and precipitation (Pmm) datasets used in this study. The mean difference and the root mean square error (RMSE) between observations and reanalysis before bias-correction/quantile mapping (bBC/QM) and after correction (aBC/QM).

| Influence | Weather station | % Missing values | Mean diff. bBC/QM | Mean diff. aBC/QM | RMSE bBC/QM | RMSE aBC/QM |
|---|---|---|---|---|---|---|
| Western | M003 T °C | 29 | 0.85 | 0.03 | 0.94 | 0.37 |
| Western | M003 Pmm | 18 | 144.58 | 1.89 | 153.57 | 54.38 |
| Eastern | M188 T °C | 45 | -1.29 | 0.33 | 1.39 | 0.61 |
| Eastern | M188 Pmm | 17 | 57.31 | 2.42 | 107.74 | 108.23 |

**3.6  Glacier clustering based on morphotopographic features**

To understand the relation between the thinning of each individual glacier and their morphometric characteristics over the entire period, we divided the glaciers into groups based on similarities in the elevation profile of annual changes in elevation. This requires a precise delineation of the boundaries of each of the glacial basins, which is difficult to achieve with medium spatial resolution imagery (*e.g.* Landsat images) of a conical structure like a volcano. However, we take advantage of the
availability of high-resolution aerial imagery. Consequently, we applied the K-means classification technique (Wu, 2012). First, each glacier was divided into regular 50-m elevation ranges, we then computed the mean annual change in surface elevation for each elevation range and these values were then randomly divided into k groups and the average for each group was computed. Next, the algorithm assigned each estimation to the closest mean value based on the Euclidean distance. The new mean value of each group was then updated and the process repeated iteratively until the sum of squares distance of the
groups was minimised. Thus, it is assumed that values in the same group underwent similar changes in their surface elevation. To define the number of clusters, we performed k-means classification for different k groups, starting with 10 groups and continuing down to one group. We then compared the sum of squares of the groups, and the breaking point in the sum of squares is an indication of the optimal number of clusters. As a result, we divided the glaciers into two groups which match with the exposure to the humid fluxes.

**4    Results and discussion**

**4.1  Decadal changes in glaciers on the Antisana ice cap**

Figure 3 gives an overview of the differences in elevation of the glaciers in the five sub-periods between 1956 and 2016. It should be noted that the five sub-periods are not representative of particular climate conditions, but are based on the availability





of remote sensing data. In all the periods, the strongest and most homogeneous changes in surface elevation occurred at lower

elevations, then faded and became heterogeneous toward higher elevations. Such noisy patterns in the upper reaches of the ice

cap (>5,100 m a.s.l.) can be attributed to the rugged topography, where seracs, crevasses and steep slopes dominate the glacial

landscape (e.g., average slope between 30° and 40°) combined with the very likely positive precipitation gradient (Basantes-

Serrano et al., 2016),  as discussed in more detail hereafter.

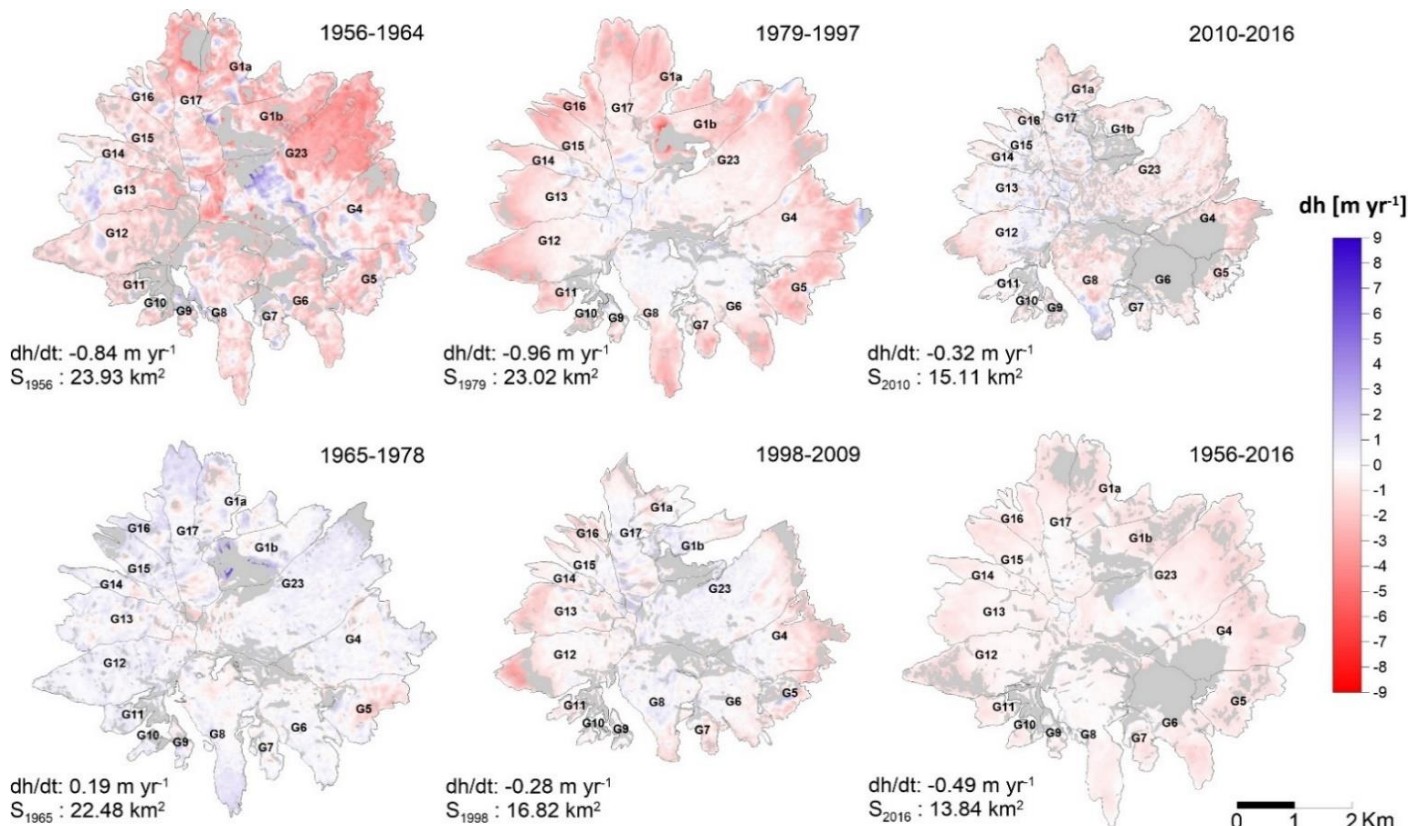

**Figure 3** Spatial variability of changes in annual surface elevation of all glaciers on the Antisana ice-cap in each study period. A decrease (resp. increase) in the glacier surface elevation is shown by the red (blue) tone scale. The glacier boundaries in the first aerial survey are indicated by the grey lines and areas for which no data are available are represented by the grey areas. The rate of elevation change and the surface area of the ice cap at first date are given on the bottom left of each map (except for the last map illustrating the entire period for which the surface area measured on the last date is given as it is not mentioned elsewhere). All the maps have the same scale.

The average elevation change for the entire period (1956-2016) was -30.01 ± 2.95 m with a minimum value of -6.58 ± 2.95 m

for glacier G9; glaciers G1a, G5, and G16 had maximum values of -54.83 ± 2.95 m, -49.50 ± 2.95 m and -58.88 ± 2.94 m

respectively. A total reduction of 42% in the surface area of the ice cap was found, from 23.93 km$^2$ in 1956 to 13.84 km$^2$ in

2016. In 2016, the average altitude of the glacier terminus ranged from 4,714 m a.s.l. to 4,794 m a.s.l. on the western slope,

and from 4,334 m a.s.l. to 4,575 m a.s.l. on the eastern slope. Values were missing for some parts of the glaciers, the 2010-

2016 period was the most affected with 26% of the surface area of the ice cap not covered by measurements, representing an

uncertainty of ± 0.20 m.





Table 4 lists the glacier-wide mass balance for all the glaciers for each sub-period. Overall, during the entire study period, glaciers underwent an average cumulative mass loss of -25.30 ± 2.51 m w.e. and an annual mass loss of -0.41 ± 0.04 m w.e. yr$^{-1}$. Glaciers G1a, G5 and G16 suffered the strongest mass losses with an average mass balance of -0.74 ± 0.04 m w.e. yr$^{-1}$.

On the contrary, glaciers G8, G9 and G10 showed moderate losses with an average annual mass balance of -0.18 ± 0.04 m w.e. yr$^{-1}$. The annual mass balances of the remaining glaciers clustered between -0.29 and -0.57 m w.e. yr$^{-1}$.

**Table 4** Glacier-wide cumulative mass balance in [m w.e.] observed during the last six decades for the 17 glaciers on the Antisana ice cap. (*) indicates periods when more than 40% of information concerning changes in the elevation of the surface area of the glacier was missing.

| Glacier | 1956-1964 | 1965-1978 | 1979-1997 | 1998-2009 | 2010-2016 | $\Sigma_{period}$ | 1956-2016 |
|---|---|---|---|---|---|---|---|
| G1a | -9.65±0.55 | 1.91±0.76 | -30.66±0.79 | -2.97±0.56 | -4.20±0.35 | -45.57 | -46.61±2.51 |
| G1b | -9.22±0.56 | 4.23±0.78 | -23.07±0.79 | 0.58±0.57 | -2.03±0.35 | -29.51 | -25.49±2.51 |
| G23 | -12.18±0.55 | 3.14±0.76 | -10.25±0.79 | 0.36±0.57 | -2.84±0.35 | -21.77 | -23.83±2.51 |
| G4 | -6.18±0.55 | 1.41±0.76 | -14.36±0.79 | -5.31±0.57 | -3.69±0.36* | -28.13 | -34.74±2.51 |
| G5 | -6.75±0.45 | -3.48±0.76 | -25.98±0.79 | -4.33±0.57 | -2.68±0.37 | -43.22 | -42.07±2.51 |
| G6 | -7.02±0.55 | 0.82±0.76 | -8.10±0.79 | -1.73±0.57 | -0.83±0.35* | -16.85 | -17.42±2.52* |
| G7 | -1.99±0.55 | 0.28±0.76 | -15.11±0.79 | -5.61±0.57 | -1.34±0.35 | -23.78 | -22.77±2.51 |
| G8 | -5.92±0.55 | 1.89±0.76 | -5.83±0.79 | 0.03±0.57 | -2.01±0.36 | -11.83 | -11.48±2.51 |
| G9 | -1.21±0.56 | 1.27±0.77 | -4.37±0.79 | -0.21±0.58* | -2.24±0.36* | -6.77 | -5.59±2.51 |
| G10 | -4.40±0.58* | 2.09±0.80* | -7.60±0.80* | -0.99±0.58* | -1.83±0.35* | -12.74 | -16.28±2.51* |
| G11 | -5.53±0.57* | 3.49±0.78 | -14.64±0.79 | -3.09±0.58 | -0.48±0.35 | -20.26 | -23.19±2.51 |
| G12 | -6.58±0.55 | 3.40±0.76 | -13.29±0.79 | -5.04±0.57 | -1.33±0.35 | -22.85 | -20.62±2.51 |
| G13 | -4.20±0.55 | 2.42±0.76 | -12.73±0.79 | -5.89±0.56 | -0.15±0.35 | -20.55 | -24.41±2.50 |
| G14 | -5.50±0.55 | 2.35±0.76 | -12.58±0.79 | -1.92±0.56 | -0.92±0.35 | -18.56 | -21.60±2.50 |
| G15 | -6.44±0.55 | 3.95±0.77 | -19.40±0.79 | -2.18±0.56 | -0.50±0.35 | -24.58 | -27.14±2.50 |
| G16 | -6.10±0.55 | 5.92±0.76 | -31.78±0.79 | -10.57±0.56 | -3.79±0.35 | -46.32 | -47.50±2.50 |
| G17 | -10.62±0.55 | 4.36±0.76 | -13.57±0.79 | 0.20±0.56 | -1.55±0.35 | -21.18 | -19.38±2.51 |
| Mean glacier wide mass balance | -6.44±0.55 | 2.32±0.77 | -15.49±0.79 | -2.86±0.57 | -1.91±0.35 | -24.38 | -25.30±2.51 |


Now to look at the different sub-periods in more detail, Antisana glaciers experienced very negative to slightly negative or even positive mass balances (Figure 4). Surprisingly, a very negative trend was observed in the 1956-1964 period, with an





average annual mass balance of -0.72 ± 0.06 m w.e yr⁻¹. During this period, the north-eastern glaciers (i.e., G17, G1a, G1b, G2,3), which represent about 40% of the total surface area of the ice cap, had the most negative mass loss rates with an average

of -1.16 ± 0.06 m w.e yr⁻¹, while the other glaciers lost mass at a two-fold less negative rate (-0.58 ± 0.06 m w.e yr⁻¹). Despite the very negative rates observed in the first period, the reduction in glacier surface area was only 8%, and remained limited until the end of the 1970s.

In contrast to the 1956-1964 period, the 1965-1978 period presented a moderate gain in ice mass by all the glaciers, with an average positive annual mass balance of 0.17 ± 0.05 m w.e yr⁻¹, which helped offset part of the mass loss that had occurred in

the previous period. Interestingly, during this period, only glacier G5 showed a mass loss of the same order of magnitude as the gains by the other glaciers. After the late 1970s, the mass balance again became negative and a long period of ice mass loss started with an annual rate of -0.82 ± 0.04 m w.e yr⁻¹. The glaciers with the most negative mass balances were G1a, G1b, G5, and G16. This overall behaviour is in line with the results of previous studies highlighting a pronounced shrinkage of tropical glaciers since the late 1970s (Masiokas et al., 2020; Rabatel et al., 2006, 2013a; Soruco et al., 2009a) At the end of the 1990s,

Antisana glaciers entered a period of limited mass loss, with an average annual mass balance of -0.26 ± 0.05 m w.e yr⁻¹ which has lasted for almost 20 years up to the present. The limited mass loss reported by (Basantes-Serrano et al., 2016) for glacier G15 was also confirmed for the other glaciers. By 2016, we observed the almost complete disintegration of G16 with a 95% reduction in its surface area since 1956.

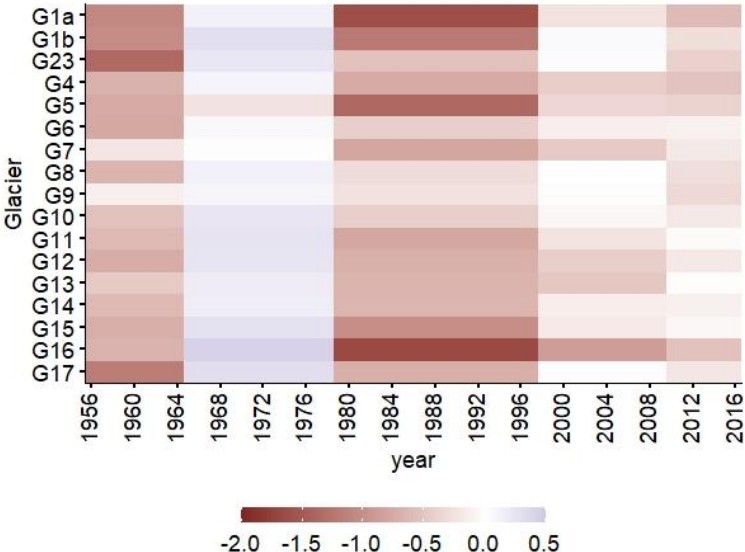

**Figure 4** Annual mass balance in m w.e. yr⁻¹ over the different periods.

## 4.2 Morphometric drivers of changes in glaciers and effects on their interpretation

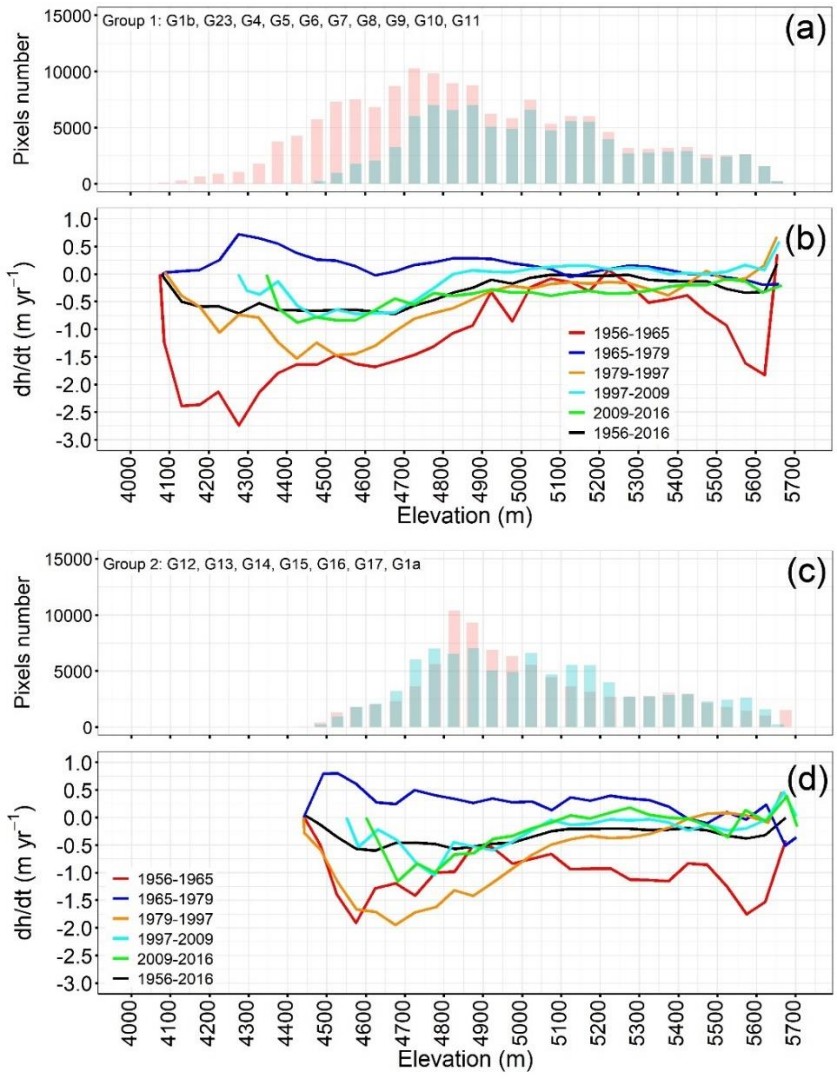

**Figure 5** Antisana hypsometry in 1956 (red bars) and 2016 (cyan bars) and the elevation change rates for the study periods as a function of the altitude of each group of glaciers: **a)** and **b)** Group I composed of glaciers exposed to Pacific fluxes, and **c)** and **d)** Group II composed of glaciers exposed to Amazon fluxes. The dh/dt values are averaged over a 50-m elevation range.

- Group I (Figure 5a), with a $B_g$ of -0.37 m w.e. yr$^{-1}$, includes very steep glaciers with slopes of more than 35°. The glacier terminus is located between 4,100 m a.s.l. (in 1956) and 4,334 m a.s.l. (in 2016). These glaciers are exposed to the east and south of the ice cap, facing the Amazon basin. Their ablation and accumulation zones are of the same width. The hypsometric changes show that the highest thinning rates occurred below the 4,900 m a.s.l, which is the mean altitude of the 0 °C isotherm.





This group includes glacier G5, which showed the highest thinning rate (-0.81±0.04 m yr$^{-1}$) of the entire period. Also, during the 2010-2016 period, an anomalous increase in the surface elevation was detected in the tongue of glacier G8. Thanks to the availability of the aerial photos taken in November 2011 by a high-resolution DMCII-143 sensor, and after visual inspection of the orthophotos in 2009 and 2011, we were able to identify a surge-type event (Meier and Post, 1969) that led an advance of about 500 m of the glacier terminus with simultaneous thickening of the glacier tongue of about 12 m (Figure 6). Indeed, above 4,800 m a.s.l. G8 has an almost flat area ($\alpha$>12°) which can be considered as a reservoir area for the storage of ice masses if the glacier receives a significant amount of solid precipitation. In the case of surge events, after some time when the storage capacity is exceeded, the material flows down into the valley to the receiving zone that then advances rapidly. In the case of G8, this mass was transferred by a "channel" with a slope of up to 22°.

Therefore, a large transfer of ice masses will appear as a drop in elevation in the departure zone (Figure 6A, arrow a) and as a rise in elevation in the lower reaches of the glacier (Figure 6A, arrow b). To our knowledge, surge events have never previously been reported in an inner-tropical glacier. Surging processes may be partially controlled by climate factors as well as by the morphology, subglacial drainage patterns, glacier hypsometry and the magnitude of previous surge events. However, at a multi-decadal time scale, the influence of climate variations tends to predominate over the influence of ice flow dynamics on glacier response (Thompson et al., 2011).

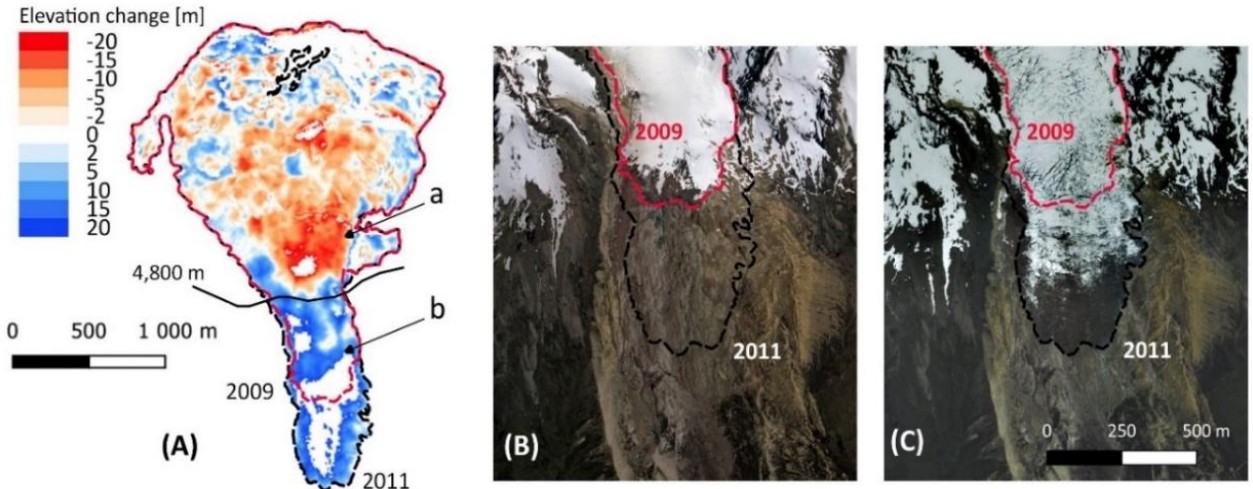

**Figure 6 (A)** Glacier G8 shows an unusual pattern of elevation changes for the period 2009-2011 (also visible for the 2010-2016 period in Fig. 3) with a decrease in elevation (red) above 4,800 m a.s.l. and an increase in elevation (blue) below 4,800 m a.s.l., according to the aerial images acquired in September 2009 **(B)** and January 2011 **(C).** The dashed lines represent the glacier outlines in September 2009 and the dashed black line represent the glacier outlines in November 2011. The 0.5 m orthophoto (2011) used as the background was provided by the Ministerio de Agricultura y Ganadería (MAG) in the framework of the "SIGTIERRAS" project.

- Group II (Figure 5b), with a $B_g$ of -0.49 m w.e. yr$^{-1}$, groups glaciers whose terminus is located between 4,550 m a.s.l. (in 1956) and 4,700 m a.s.l. (in 2016). These fan-shaped glaciers have a similar morphology with gentler slopes





(5°<$\alpha$<15°) in the wider lower parts and steeper topography ($\alpha$>35°) in the upper narrower zones. The change in surface area of these glaciers occurred between 4,800 and 5,000 m a.s.l, whereas upslope the change in the surface elevation of these glaciers was very limited. In this group, the average thinning of glaciers G1a and G16 (-0.91±0.04 m yr$^{-1}$) was strong over the entire period.

To understand the sensitivity of the glaciers to the morpho-topographic features in more detail, we performed a regression analysis between the average glacier-wide annual mass balance for the 1956-2016 period and morpho-topographic factors ( Table 5). The balances for the entire study period were moderate but significantly correlated with the following variables: (i) maximum altitude; (ii) median altitude, a variable considered as a proxy of the balanced-budget ELA (Braithwaite and Raper, 2009; Rabatel et al., 2013a); (iii) mean slope; (iv) the cosine of the mean exposure; and v) the potential of incoming solar

radiation under clear-sky conditions (Fu and Rich, 2003; Rich et al., 1994). The morpho-topographic characteristics of the glaciers explained 30-40% of the observed long-term variation in the mass balance of the glaciers. However, when we considered the glaciers based on their group, the relationship between the mass balance and the maximum and median altitude, and solar radiation was largely improved.

We observed that slope played an important role in defining the rate of ice losses of the glaciers in Group I. Glaciers with a

mean slope of less than 30° and that extended above 5,300 m a.s.l. presented a moderate imbalance. In fact, a relatively small ice loss is expected in steep slope sectors (Venkatesh et al., 2012) as long as the ice masses are present at altitudes where the conditions favour continued accumulation (see Table S1 in supplementary materials). Also, around 60% of the variance of the mass balance can be explained by solar radiation. When all the glaciers were included, we found a significant correlation between the mass balance and changes in surface area, and when the very unbalanced glaciers (G1a, G5 and G16) were

removed, we found a lower but nevertheless significant correlation between mass balance and changes in surface area. The higher explained variance when the very unbalanced glaciers were included may be due to the morphometrical characteristics of these glaciers, whose maximum altitude is below 5,300 m a.s.l., with no accumulation area to balance the high rates of mass losses in the lower reaches due to a flux of mass from the upper reaches, thus leading to a bigger reduction in surface area.

**Table 5** Correlation coefficient $R^2$ for the regression analysis between the conventional and the morpho-topographic features for the
period 1956-2016. The regression analysis covered all the glaciers $Bm$ and the $'Bm$ but not the outlier glaciers (G1a, G5 and G16). The level of significance $p$-value < 0.01(***), $p$-value < 0.05 (**), and $p$-value < 0.10 (*).

| Variables | $R^2 - Bm$ All glaciers | $R^2 - 'Bm$ All glaciers | $R^2 - Bm$ Group 1 | $R^2 - Bm$ Group 2 | $R^2 - 'Bm$ Group 1 | $R^2 - 'Bm$ Group 2 |
|---|---|---|---|---|---|---|
| Maximum altitude | 0.30** | - | - | 0.76*** | - | 0.60** |
| Median altitude | 0.40*** | 0.38*** | 0.89*** | 0.67** | 0.86*** | 0.55** |
| Mean slope | 0.28** | 0.45*** | 0.47** | - | 0.60*** | - |
| Exposure | 0.34*** | 0.42** | 0.29* | - | 0.43** | - |
| Solar radiation | 0.17* | 0.32*** | 0.47** | 0.62** | 0.60*** | 0.57** |
| Surface area changes | 0.80*** | 0.48*** | 0.65*** | 0.98*** | 0.44** | 0.84*** |





To test the sensitivity of the glacier mass balance to variations in glacier surface area, we computed the glacier-wide mass balance for each sub-period by keeping the glacier surface area unchanged (hereafter glacier-wide fixed mass balance $B_{g_f}$).

We thus replaced the term $\overline{S}$ in Eq. (5) by $S_o$ the surface area of the glacier at the date of the first geodetic survey in the study sub-period concerned. Therefore, one would expect the mass balance of the entire ice cap to be overestimated if the mean variation of the surface area of the small glaciers due to their imbalance with climate is not taken into consideration.

The cumulative mass balance of the whole ice cap over the entire period resulted in a discrepancy of 3.6 m w.e. between the $\sum B_g$ and $\sum B_{g_f}$, which means that $B_{g_f}$ is 0.11 m w.e. yr$^{-1}$ less negative than $B_g$ (-0.42 m w.e. yr$^{-1}$). Considering the glaciers

according to their group, the difference between $B_g$ and $B_{g_f}$ was 0.09 m w.e yr$^{-1}$ for glaciers in Group I, which falls within the uncertainty of the mass balance quantification (Figure 7a). This was not the case for glaciers in Group II where the difference between $B_g$ and $B_{g_f}$ was 0.15 m w.e. yr$^{-1}$ (Figure 7c). At the scale of the sub-periods, the biggest differences between the two balances occurred in the 1979-1997 period with an overestimation of 0.12 m w.e. yr$^{-1}$ for Group I and of 0.20 m w.e. yr$^{-1}$ for Group II (Figure 7b and d). For the other sub-periods, the difference is within the uncertainty of the average mass balance.

When glaciers G5 (Group 1) and G1a and G16 (Group 2) were excluded, the discrepancy between the $B_g$ and $B_{g_f}$ was reduced (Figure 7e to h) showing that: (i) up to 30% of the overestimation in the mass balance becomes apparent if the mean variation in the surface area of the ice cap is disregarded; and (ii) most of this discrepancy may be due to a marked imbalance between the geometry of the small glaciers and the climate.

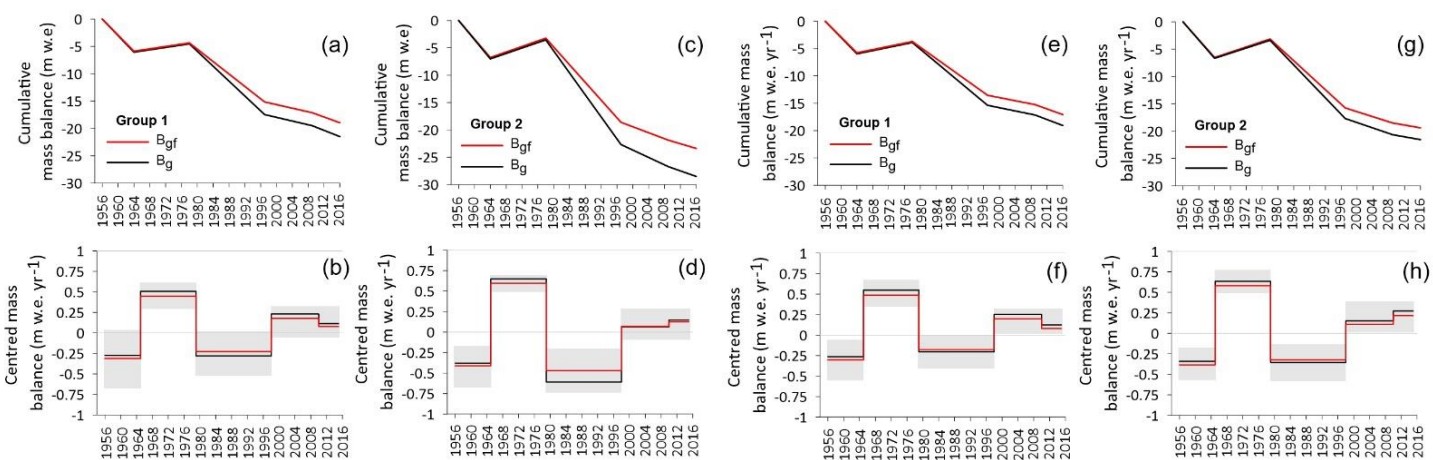

**Figure 7** Cumulative mass balance and centred mass balance of the glaciers in Group I (including glacier G5 in a and b, and excluding G5 in e and f) and of the glaciers in Group II (including G1a and G16 in c and d, and excluding G1A and G16 in g and h). The black line represents the conventional balance and the red line represents the fixed mass balance. The grey shaded areas represent the standard deviation of the mass balance.

Thanks to the availability of recent geodetic observations provided by satellites, several studies have been undertaken to

estimate the glacier mass loss at regional scale in recent decades (e.g., Braun et al., 2019; Brun et al., 2017; Dussaillant et al., 2019). Because no multitemporal inventory is available to quantify changes in the glacier surface area in the majority of the



glacierised regions in the world, many regional studies assume the glacier surface area remains constant. As underlined by Dussaillant et al., (2019) at regional or mountain range scale, the impacts of this assumption are limited, but at glacier scale and particularly in periods with significant glacier retreat, not accounting for changes in the surface area of the glacier may
result in underestimation of ice mass loss as is shown in this study. This inaccurate estimation was as high as 30% in the case of the smaller glaciers of the Antisana ice sheet studied here. (Fig. S2 in supplementary materials). This fact is extremely important, for example, at local scale, when assessing the contribution of glacier melt to the catchments that supply water to local communities.

### 4.3 Glacier response to climate variability

To analyse the decadal variability of glacier volume change, we included $B_g$ but not glaciers with a strong geometric effect, and we computed the mean centred mass balances of the glaciers for each sub-period by subtracting the average mass balance for the 1956–2016 period. This resulted in centred mass balance values of -0.29 m w.e. yr$^{-1}$ ($\sigma=\pm0.33$), 0.58 m w.e. yr$^{-1}$ ($\sigma=\pm0.09$), -0.26 m w.e. yr$^{-1}$ ($\sigma=\pm0.27$), 0.22 m w.e. yr$^{-1}$ ($\sigma=\pm0.20$), 0.18 m w.e. yr$^{-1}$ ($\sigma=\pm0.14$), for 1956-1964, 1965-1978, 1979-1997, 1998-2009 and 2010-2016, respectively. Periods with negative mean centred mass balance values (i.e., negative
anomalies) showed more dispersion between glaciers than periods with positive anomalies. However, the overall picture shows a homogeneous response of glaciers to climate variability. Although the periods for which the mass balance variability was reconstructed based on the available images using the geodetic method are constrained by the dates of the surveys, the mass balance reflects the decadal climate conditions and/or changes.

To link the regional climate to the decadal variability of the mass balance, we compared the centred geodetic mass balance
with climate anomalies computed from ERA5 reanalysis data, *i.e.,* anomalies in temperature, and in specific humidity as a proxy of precipitation anomalies (Fig. 9) (Ruiz-Hernández et al., 2021). To be able to see the footprint that ENSO events may have left on the decadal mass balances, we included in the comparison the Southern Oscillation Index (SOI) averaged for each sub-period (Ropelewski and Jones, 1987). SOI is a standardized index based on changes in ocean temperatures across the eastern tropical Pacific during ENSO events. Negative values correspond to warm conditions during "El Niño", while positive
values correspond to cold temperatures during "La Niña" episodes. To understand the effect of seasonal variability in the long-term glacier response, we focused our analysis on the key quarters of the year, MAM and SON (see Appendix S1 in supplementary materials). Anomalies were determined relative to the current normal climate period 1981-2010. For instance, a year is considered to be anomalous if temperature anomalies are greater than ±0.5°C and precipitation anomalies are less than ±10%. Likewise, as a proxy of the energy available for melting, we included the estimated average anomaly of downward
solar radiation (srad) per period taken from the ERA5 reanalysis as a proxy of shortwave radiation at the Earth's surface. It is worth mentioning that this is a broad approximation based on the average solar radiation over a grid box on the ice cap.




**Figure 8** Climate anomalies relative to the current normal climate period 1981-2010 in **a)** temperature and **b)** specific humidity at pressure levels over Antisana ice cap. **c)** and **d)** the same as in a and b, but for the MAM quarter, and **e)** and **f)** for SON quarter. Pale to dark red represent warm and dry, whereas pale to dark blue represent cold and humid. The dashed line shows the maximum altitude of the Antisana ice cap. **g)** The centred mass balance for Antisana ice cap (black line) and the (1σ) standard deviation (grey shaded area) for each sub-period, the orange (green) line represents SOI fluctuation averaged over the MAM (SON) quarter. **h)** The centred mass balance compared with the 11-month moving average of the downward solar radiation anomaly over the Antisana ice cap (red line).

The main patterns that emerge from Figure 8 are described below:

– In the 1956-1964 period, a notable loss of ice mass was observed which can be attributed to the dry conditions that prevailed in the 1950s and were further accentuated in the 1960s. Dry conditions were even more marked in the SON quarter. Regarding temperature, this period was characterized by a negative anomaly (ΔT < -0.2 °C compared to the mean calculated over the period 1981-2010). Thus, while dry conditions may have limited accumulation in the upper reaches of the ice cap, a concomitant reduction in cloudiness may have been responsible for an increase in shortwave radiation

(Figure 8h) thereby enhancing melt due to the decrease in albedo. Sporadic snowfall in the MAM quarter may not have been sufficient to reduce the steep ablation rates.



- In the 1965-1978 period, the glacier mass balance was stable or even positive. These conditions are consistent with negative temperature anomalies and a mostly positive humidity anomaly which likely triggered snowfall all year round, thereby increasing glacier surface albedo, reducing ablation, and enhancing accumulation processes over the ice cap. Later,

in a colder and more humid context, the probable increase in cloudiness could have reduced the energy available for melt (negative solar radiation anomaly Figure 8h) and snowfall events would continue to enhance accumulation processes until the mid-1970s. In the late 1970s, slightly warm and moderately dry conditions emerged which started the long negative period for the ice cap described below.

- In the 1979-1997 period, strong ice mass losses predominated. Warm conditions prevailed, and were slightly more marked

during the key quarters ($\Delta T > +0.25$ °C). These conditions were combined with a long, slightly wet period but which was interrupted by some dry years. Particular features were: (i) A warm and humid context during MAM in the 1990s provided favourable conditions for rainfall on the tongues of the glaciers thereby increasing melt rates. A very strong "El Niño" occurred in 1983 characterised by a positive temperature anomaly combined with very humid conditions favouring rainfall on the lower reaches of the glaciers, which would increase ablation rates via the shortwave radiation budget due to the

low albedo. (ii) After 1985, warm conditions combined with the surplus humidity recorded in the SON quarter, enhanced melt rates, which were further increased during the strong "El Niño" in 1987. The observed slight increase in incoming shortwave radiation may have helped maintain high melt rates.

- In the period following the late 1990s - early 2000s, slightly negative or even positive mass balances were documented. In humid conditions, presumably more continuous cloudiness over the volcano helped reduce shortwave radiation at the

glacier surface. On the other hand, precipitation and slightly colder episodes could have maintained the snow cover long enough to protect the glaciers from the energy available for melting. One strong cold "La Niña" event occurred in 1999/2000 along with two warm "El Niño" events, one in in 1998 and the other in 2015/2016. After 2015, conditions suddenly changed to become moderately warm and humid over the glaciers increasing rainfall rather than snowfall and reducing the albedo, hence increasing ablation rates.

- The consistency between the decadal variability of the mass balance, the variability of the shortwave radiation available for melting, and the large-scale fluctuations in the sea surface temperatures between the western and eastern tropical Pacific represented by the SOI index is clearly visible throughout the study period (Figure 8g).

## 5   Conclusion

This paper has provided a detailed view of the changes to glaciers in the Antisana ice cap, Ecuador, i.e., in the inner tropics,

since the middle of the 20$^{th}$ century. Despite the contrasted decadal variability of the mass balance, an overall negative trend prevailed over the last seven decades with an annual mass loss rate of -0.41 ± 0.04 m w.e. yr$^{-1}$. This study revealed that glacier mass losses were remarkably high (-0.72 ± 0.06 m w.e yr$^{-1}$) in the 1956-1964 period, whereas a moderate gain in ice mass was detected in the 1965-1978 period (0.17 ± 0.05 m w.e yr$^{-1}$) which partially offset glacier shrinkage in the preceding period. After the late 1970s, ice mass loss continued until the late 1990s (-0.82 ± 0.04 m w.e yr$^{-1}$). However, since the early 2000s,





thinning of the glaciers of the Antisana ice cap has slowed down, as revealed by slightly negative mass balances (-0.26 ± 0.05 m w.e yr$^{-1}$). The overall change in the mass of the ice cap between 1956 and 2016 reduced its surface area by 42%. Glaciers G12 and G13, located on the western side, roughly followed the same general pattern observed in each sub-period, confirming that these glaciers are fairly representative of the decadal glacier response, and could thus be used as benchmark glaciers for future studies.

Overall, 70% of the thinning occurred below the average elevation of 0°C isotherm (around 4,900 m a.s.l.). We distinguished two types of glaciers whose response to climate variability is determined by their morpho-topographic characteristics. The spatial pattern of the changes in glacier surface elevation is more marked on the eastern slopes of the ice cap where the glaciers are more directly influenced by the interplay between the flows of humidity from Amazonia and the rugged and heterogeneous ice cap morphology. Interestingly, a surge-type event was detected on glacier G8 underlining the importance of including the

effect of ice-flow dynamics as well as climate variations when studying glacier response. To our knowledge, no similar event has been reported in the tropics to date, thus more research is needed before being able to conclude on the internal (ice-flow dynamics) or external factors (climate) that triggered such an event.

Regression analysis between mass balance rates over the whole period and the morpho-topographic variables, suggests that about 40% of the mass balance spatial variability can be explained by the topography. However, glaciers located on the western

side reacted more obviously to topographic features. We found that glaciers whose terminus reached 4,100 m a.s.l. in 1956 retreated further than glaciers whose terminus was located at 4,700 m a.s.l. Furthermore, small glaciers with a maximum elevation below 5,300 m a.s.l. have almost disappeared (~86% loss of surface area since 1956) as reported for other glaciers in the region, and the remaining portions of ice will probably disappear within the coming decade. Additionally, our study shows that estimations of geodetic mass balance are prone to error when variations in glacier surface areas are neglected,

particularly at the glacier scale and in the case of small glaciers when the loss is high. This can indeed lead to a misinterpretation of the glacier response to climate and may have a major impact on hydrological modelling used to estimate the contribution of meltwater in catchments that contain small glaciers.

Overall, the behaviour of glaciers on the Antisana ice cap is consistent with the regional climate signal, which shows a stepwise transition towards warmer and alternating wet/dry conditions since the mid-1970s. Sub-periods with extreme mass balance

anomalies have more scattered precipitation values than years with moderate or even slight mass balance anomalies.

Our results provide a starting point for future research on the physical processes that drive these changes in the complex topographical-climatic conditions found in the tropical Andes. Due to the importance of this site as a water resource for the population of the city of Quito, we also recommend analysis of the contribution of meltwater to the hydrological budget of the catchments, which is crucial for effective management of water resources under climate change.

**Code and data availability**

The changes in annual surface elevation of all glaciers on the Antisana ice-cap in each study period produced by this research, can be found and downloaded from https://glacioclim.osug.fr/-Acces-a-des-donnees-elaborees-.



**Author contribution**

RBS designed the study, produce the datasets, led the geodetic and climate analysis, and wrote the manuscript. AR, BF
designed the study, and contribute to writing the manuscript. CV provided recommendation on the glaciological and geodetic
interpretations. AS provided guidance on the photogrammetric processing. TC and JCR contribute with climatic interpretation.
All authors contributed to the data interpretation and participated in discussions for improving the manuscript.

**Competing interests**

The contact author has declared that neither he nor his co-authors have any competing interests.

**Acknowledgments**

This study was funded by *Universidad Regional Amazónica Ikiam* and the *Laboratoire Mixte International* GREAT-ICE
(http://www.great-ice.ird.fr/) lead by the French Institute of Research for Development (IRD). We acknowledge the *Service
National d'Observation* GLACIOCLIM (http://glacioclim.osug.fr/) and the contribution of the Labex OSUG@2020
(*Investissements d'Avenir* – ANR10 LABX56). Special thanks to the institutions that provided access to the aerial photographs
and meteorological observations: the "*Ministerio del Ambiente del Ecuador*" (MAE), the "*Empresa Metropolitana de
Alcantarillado y Agua Potable de Quito*" (EPMAPS) and the "*Instituto Nacional de Meteorología e Hidrología*" (INAMHI).
We acknowledge Dr. Etienne Berthier (CNRS, LEGOS) for the Pleiades Glacier Observatory initiative via the French Space
Agency's (CNES) ISIS programme which facilitated access to Pleiades data. ERA5 Reanalysis data were acquired through the
Copernicus Climate Change Service implemented by the European Centre for Medium-Range Weather Forecasts (ECMWF).

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
