# Peer review of "New insights into the decadal variability in glacier volume of a tropical ice-cap explained by the morpho-topographic and climatic context, Antisana, (0°29' S, 78°09' W)"

_The Cryosphere, 2022_

## Author Comment (AC1)

Dear Reviewer

We deeply thank you for your reading and the remarks provided on the manuscripts. All your comments have been considered and the references have been included. Hereafter, you will find our answers to your questions (in green).

Best regards,

**Review 1** to Basantes-Serano et al. (2022): "New insights into the decadal variability in glacier volume of an iconic tropical ice cap explained by the morpho-climatic context, Antisana (0°29' S, 78°09' W)" submitted to The Cryosphere.

The authors present a study of the photogrammetrically derived surface elevation changes of the Antisana ice cap. The surface elevation changes since the mid-20$^{th}$ century are separated into five subsets, each roughly representing a decade, converted into mass changes, and correlated to their morpho-climatic setting.

Overall, this manuscript is well structured and presents a very sound study. The research questions are laid out clearly, the methods are explained and applied properly. The significance of the study is high, because it extends our knowledge of one of the best studied glaciers in the tropics, a region where long-term glacier observations are particularly rare but highly demanded for calibration of glaciological and hydrological models.

I have some comments to the text, datasets and methods, which I assume to be a major revision.

We thank for your positive and useful remarks. In the new version, the manuscript has been updated following your comments.

My main points of concern are:

The coverage of dh-samples (L 178-184): I suggest adding a figure (supplement) showing the data coverage across the glacier hypsometry. I just wonder if the coverage is evenly distributed or if certain elevation bands or larger areas are systematically missing and the authors possibly introduce a bias.

We check for that possibility of systematic bias, and we confirm that all the dh-samples are randomly distributed over the glacier. A new figure has been added in supplementary materials (Fig. S2). In the sake of clarity, you can now read in the Uncertainty Analysis section (point fourth): *"In addition, it is worth mentioning that the dh coverage for all periods are evenly distributed over the glacier surface, which reduces the likelihood of inducing some spatial biases in the quantification of glacier elevation changes (Fig. S2 in supplementary materials)."*

The density used to drive elevation changes into mas changes (eq. 5). I'd like the authors to elaborate a bit more on the density used. I know, it is convenient and mostly enough to use the approach of Huss (2013), especially in periods of mass loss. However, the second period in this study (1965-1978) is characterized by mass gain and this raises the question of possible lower densities than 850 kg/m³, because it needs several decades to compact snow to ice. It would be worth to look into possible other sources of density (e.g. Williams et al., 2002).

Reply R1: We agree with this comment. The new version of the manuscript considers two density assumptions (see Uncertainty analysis section). One is a conservative scenario recommended by Huss (2013) for the periods of mass losses and a second scenario is based on observational data from Williams et al. (2002) for the periods of mass gains.

You can now read: *"Second, regarding the uncertainty related to the density assumption, we analyze two extreme scenarios: First, we consider an average density recommended by Huss (2013) of $\bar{\rho}$ = 850 kg m$^{-3}$ with a plausible uncertainty range of $\sigma_\rho$ = ±60 kg m$^{-3}$. This value is appropriate for a wide range of conditions and when no information on firn pack changes is available (Huss, 2013; Zemp et al., 2013). However, moderate mass gains occurred in the second study period for which the conventional density assumption may not be true. Taking advantage of firn compaction data in two shallow core (mean depth ~14m) extracted from the summit of the Antisana volcano in February 1996 and November 1999, respectively (Calero et al., 2022; Williams et al., 2002), we propose a second scenario with an average density value of $\bar{\rho}$ = 564 ± 64 kg m$^{-3}$, indicating that the mass gain or loss was mostly comprising firn."*

We also include a section to discuss the sensitivity of the mass balance to the density assumption. Now you can read as follow:

*"4.2    Sensitivity of the geodetic mass balance to the density assumption*

*In most of the geodetic studies, when there is no information available about changes in firn pack it is strongly recommended to use a conservative density value such as the one proposed by Huss (2013), especially in periods of mass loss. However, in our glaciers, the second period (1965-1978) is characterized by mass gain and a density value close to the density of ice could led to an overestimation of the mass balance. Assuming a density of 850 kg m$^{-3}$ both in the accumulation and ablation areas for 1965-1978 period, the mass balance increase to 0.06 m w.e yr$^{-1}$ which is within the uncertainty of the mass balance. In addition, during 1998-2009 period, seven glaciers in the Antisana ice cap are close to equilibrium with a slightly positive or negative mass balance no matter what density scenario is assumed. Given the small difference between both assumptions, we decide to apply an average density value of 850 kg m$^{-3}$ when mass losses prevails, and when positive conditions are present we use an average density of 564 kg m$^{-3}$ according observational data in the summit of the Antisana ice cap (Calero et al., 2022; Williams et al., 2002)."*

L 99: At what altitude is the maximum precipitation rate recorded? On tropical mountains the positive vertical precipitation gradient often reverses above a certain altitude. Is this the case for Antisana as well?

Gridded precipitation: It would be good to have some explanation why ERA5 precipitation was selected for the analyses. There are other gridded precipitation data sets like the PSL South America daily gridded precipitation, GPCC, or a recently published data set for the Peruvian and Ecuadorian water sheds (Fernandez-Palomino et al., 2021). Ideally, the authors add a figure (supplement) relating stations M003 and M188 to the gridded data. For example, show in two panels the time series of monthly air temperature and precipitation from a gridded data set (monthly box plots or shading ±1 standard deviation from a reference period) and the station measurements as lines.

The highest rate of precipitation (3,700 mm yr-1) is at 3,800 m asl in the western side of the volcano, to 7 km away from the ice cap (see Figure 1 in Ruiz-Hernández et al., (2021)). Based on ERA5 reanalysis Ruiz-Hernández et al. (2021) analyze the circulation patterns linked to the precipitation variability at 86 stations in the surrounding of the Antisana volcano. Authors shows a strong spatial variability of precipitation between the western and the eastern side of

the cordillera. Note that Antisana ice cap is a transition zone between regions under Pacific influence and regions controlled by the atmospheric processes of the Amazonian basin. Thus, the annual precipitation in the inter-Andean valley varies from 400 mm yr$^{-1}$ to 1500 mm yr$^{-1}$. Instead, the headwater on the Napo River displays some values of annual precipitation that may reach up to 6000 mm yr-1 on Amazon foothills. Also, the authors report a positive annual gradient of precipitation of 200 mm per km of altitude on the inter-Andean valley, which is in line with a hypothesis proposed by Basantes-Serrano et al. (2016) that could explain almost balanced conditions on the Antisana 15α Glacier for 1995-2012 period.

Consequently, ERA5 database was used since this new reanalysis benefits from several improvements in numerical weather prediction based on vast amounts of historical measurements. Likewise, ERA5 reanalysis allows us to interpret the climate behavior in a wide range of pressure levels over the region (see Figure 8). Moreover, ERA5 has a much higher temporal (since 1950 at daily time step) and spatial resolution (30 km) than previous global reanalysis.

The additional dataset mentioned by the reviewer are not suitable for the focus of this study. The following are some of the limitations:

1.  RAIN4PE dataset was published by Fernandez-Palomino et al. (2022) as a result of the combination of precipitation data including ERA5 reanalysis among others data sources. Although RAIN4PE has a high spatial resolution (10x10 km), it covers only a part of our study period i.e., from 1981 to 2015.

2.  Unfortunately, we did not have access to the PSL South America daily gridded precipitation dataset because the website is broken https://psl.noaa.gov/data/gridded/data.south_america_precip.html .

Now, you can read: *"This data set was selected because it well represents the long-term climatology of this region (Fig. S3 in supplementary materials), moreover it covers several geopotential levels and the entire study period."*

Comments:

Title: I suggest rewording the title omitting "new insights" and "iconic". The first, because apart from the surge of G8 I speculate nothing is really new, and I assume most results are an upscaling or a validation of assumptions based on earlier studies of G15 or other tropical glaciers. The latter, because it reads more lurid than scientific.

We agree partially with the reviewer, although our extended dataset allows to confirm a synchronous response between the glacier mass changes and climate in the long term, we consider there are some original results as the almost balanced conditions observed in the second period; a differential response between eastern and western glaciers, and of course the surge event in G8.

The title was adjusted, you can now read: *"New insights into the decadal variability in glacier volume of a tropical ice-cap explained by the morpho-topographic and climatic context, Antisana, (0°29' S, 78°09' W)"*

L 43: Here you could connect to Nicholson et al. (2013), who compare the micrometeorological conditions of small tropical glaciers.

Ok, the reference is now included.

L 154: Could you explain in a half sentence why the geometric characteristics were optimal?

The sentence was a bit confusing. In fact, it is better to mention the photogrammetric adjustment instead of geometric characteristics. You can now read: *"As the 2009 geodetic survey presents the best the photogrammetric adjustment…"*

L 180: Could you explain this spatial optimisation?

The geostatistical framework for spatial optimization of dh coverage is described in detail in Basantes-Serrano et al. (2018). Nevertheless, the procedure is described focusing on the sample design, the amount of measurements, and the type of model used to adjust the spatial structure of the dh. We encourage readers to review the geostatistical procedure in depth by referring to the article devoted to this approach. However, the new version of the paper includes a brief description of the method. Now, you can read: *"[…] relies upon the spatial variability of the elevation change to densify a sampling network to optimize the quantification of the surface-elevation change."*

L 206: I assume Scor needs a capital S.

Ok, corrected

L 248: I think you should state how many ERA5 grid cells cover your domain (Fig. 2).

For the analysis about the climate drivers, we use a grid cell over the station location, this is included in the text L 257. We believe it is not necessary to mention again to avoid redundancy.

L 313: termini (plural)

Ok, adjusted

L 350 and whole chapter 4.2. I think the two groups from Fig. 5 are confused with the two groups in the text. The figure caption says group I are the glaciers at the Pacific side. In the text it is the other way round. Please correct.

Ok, now the text in the caption agrees with the text in the description.

L 364: Is this mass gain (1998-2009) also detectable in the precipitation time series? (Another reason to add a figure about precipitation).

Figure 8 shows a positive anomaly of specific humidity that begins in the late 1990s and strengthens during the last few years. This anomaly is coincident with an increase in temperatures. This is mentioned in the text: *"In the period following the late 1990s - early 2000s, slightly negative or even positive mass balances were documented. In humid conditions, presumably more continuous cloudiness over the volcano helped reduce shortwave radiation at the glacier surface. On the other hand, precipitation and slightly colder episodes could have maintained the snow cover long enough to protect the glaciers from the energy available for melting."* We also mention a plausible effect under the presence of ENSO events during this period; SOI index displayed more frequent cold conditions which could contribute to reduce mass losses.

L 369: Please, elaborate on the role of subsurface heating as possible reason for the surge. L78 suggests that Antisana is an active volcano.

Although the volcano is considered to be active, there is no indication of volcanic activity over the past 400 years (personal communication from ML Hall, 2014). The volcano has been considered as a dormant volcano for more than a century and there is no evidence for geothermal activity, surface deformation or a local decrease in ice due to hot streams on the glaciers and the surrounding terrains (personal communication from P Ramon, 2014). Nevertheless, we cannot reject with 100% of certainty that the surge event is not related to an increase in basal melt due to heat transfer from the volcano. Unfortunately, heat fluxes have not been measured to confirm this hypothesis. If geothermal contribution exist, this would be very local.

Following your comment, we added this hypothesis in the manuscript:

- In section 4.3: "*In the present case, it could be hypothesized that sub-surface heating enhancing basal melt might be part of the triggers of this surge event, but no volcanic activity has been evidenced over the past four centuries*."
- In the Conclusion: "To our knowledge, no similar event has been reported in the tropics to date, thus more research is needed before being able to conclude on the internal (ice-flow dynamics) or external factors (climate, *surb-surface heating due to volcanic activity*) that triggered such an event."

L 371/372: This is an odd sentence and consider deleting it. I even doubt the message is true, because the ice flow dynamics are a consequence of the climate variations, and especially at longer time scales glacier response times will be met. Then ice flow dynamics reflect climate variations. The reference (Thompson) is missing in the reference list.

Ok, the sentence was deleted.

L 400: This sentence is difficult to read. Consider restructuring.

In the sake of clarity the text was adjusted, you can now read: *"The higher explained variance when the very unbalanced glaciers were included may be due to the morphometrical characteristics of these glaciers, whose maximum altitude is below 5,300 m a.s.l., leaving a wider ablation area than the accumulation area, thus leading to a bigger reduction in surface area."*

Table 5: Explain Bm and 'Bm.

Ok, the text was edited to explain those variables *"Considering the mass balance for all the glaciers ($Bm$) of the ice cap and excluding the outlier glaciers (G1a, G5 and G16) ($'Bm$)."*

L 409: I suggest adding a few sentences on the concept of the reference surface balance (Elsberg et al., 2001; Harrison et al., 2005; Huss et al., 2012).

Ok, references were added concerning the reference mass balance: *"This exercise was inspired by the work on the reference mass balance at annual time step conducted by Elsberg et al. (2001); Harrison et al. (2005) and Huss et al. (2012), which were focused on the effect of climate in the glacier response without taking into account the change in geometry due to flow dynamics."*

L411: I don't fully understand why the mass balance should be overestimated in this case. When referring to a larger surface area the specific mass balance should become a smaller

number, thus an underestimation as shown in L 414 and Fig. 7. Same problem in L 435 (underestimation). Maybe a clearer wording does the job.

Sorry for this mistake. We edited the text according your suggestion.

L 432: … many regional studies: Please add the references.

Ok, references are included: *(Braun et al., 2019; Dussaillant et al., 2019)*

L 470 et seq. and Fig. 8: Very interesting figure. Consider adding a note, that tropical glaciers are known to be particular sensitive on moisture/precipitation/clouds (Mölg et al., 2009; Prinz et al., 2016; Sicart et al., 2005).

In the sake of clarity the text was adjusted. You can now read: *"As it is well known glaciers in this region are particularly sensitive to humidity and as a consequence to precipitation and clouds (Mölg et al., 2009; Prinz et al., 2016; Sicart et al., 2005),"*

L 680: Consider deleting Hastenrath 1981.

Ok, adjusted

References:

Elsberg, D. H., Harrison, W. D., Echelmeyer, K. A. and Krimmel, R. M.: Quantifying the effects of climate and surface change on glacier mass balance, J. Glaciol., 47(159), 649–658, doi:10.3189/172756501781831783, 2001.

Fernandez-Palomino, C. A., Hattermann, F. F., Krysanova, V., Lobanova, A., Vega-Jácome, F., Lavado, W., Santini, W., Aybar, C. and Bronstert, A.: A novel high-resolution gridded precipitation dataset for Peruvian and Ecuadorian watersheds – development and hydrological evaluation, J. Hydrometeorol., 23(3), 309–336, doi:10.1175/JHM-D-20-0285.1, 2021.

Harrison, W. D., Elsberg, D. H., Cox, L. H. and March, R. S.: Different mass balances for climatic and hydrologic applications, J. Glaciol., 51(172), 176, 2005.

Huss, M.: Density assumptions for converting geodetic glacier volume change to mass change, Cryosph., 7(3), 877–887, doi:10.5194/tc-7-877-2013, 2013.

Huss, M., Hock, R., Bauder, A. and Funk, M.: Conventional versus reference-surface mass balance, J. Glaciol., 58(208), 278–286, doi:10.3189/2012JoG11J216, 2012.

Mölg, T., Cullen, N. J., Hardy, D. R., Winkler, M. and Kaser, G.: Quantifying climate change in the tropical midtroposphere over East Africa from glacier shrinkage on Kilimanjaro, J. Clim., 22(15), 4162–4181, doi:10.1175/2009JCLI2954.1, 2009.

Nicholson, L. I., Prinz, R., Mölg, T. and Kaser, G.: Micrometeorological conditions and surface mass and energy fluxes on Lewis Glacier, Mt Kenya, in relation to other tropical glaciers, Cryosph., 7(4), 1205–1225, doi:10.5194/tc-7-1205-2013, 2013.

Prinz, R., Nicholson, L., Mölg, T., Gurgiser, W. and Kaser, G.: Climatic controls and climate proxy potential of Lewis Glacier, Mt. Kenya, Cryosph., 10(1), 133–148, doi:10.5194/tc-10-133-2016, 2016.

Sicart, J. E., Wagnon, P. and Ribstein, P.: Atmospheric controls of the heat balance of Zongo Glacier (16°S, Bolivia), J. Geophys. Res., 110(D12), D12106, doi:10.1029/2004JD005732, 2005.

Williams, M. W., Francou, B., Hood, E. and Vaughn, B.: Interpreting Climate Signals from a Shallow Equatorial Core: Antisana, Ecuador, in The Patagonian Ice Fields, edited by G. Casassa, F. V. Sepulveda, and R. M. Sinclair, pp. 169–175, Kluwer Academic / Plenum Publishers, New York., 2002.

---

## Author Comment (AC2)

Dear Reviewer

We deeply thank you for your reading and the remarks provided on the manuscripts. All your comments have been considered and the references have been included. Hereafter, you will find our answers to your questions (in green).

Best regards,

**Review2** of "New insights into the decadal variability in glacier volume of an iconic tropical ice-cap explained by the morpho-climatic context, Antisana, (0°29' S, 78°09' W)" by Basantes-Serrano et al., (2022).

**General comments**

This article describes the decadal changes in glacier volume in the Antisana ice cap located in the tropical Andes, Ecuador. The authors have used photogrammetric and remote sensing techniques to provide a long-term geodetic mass balance for the Antisana ice cap. Overall, there has been a lack of long-term glacier mass balances studies in this region. For this reason, additional information and novel insights into the past and current state of tropical glaciers are very welcome. In general, I think this is a well-presented and worthwhile piece of research and could help increase our knowledge about the spatiotemporal patterns of glacier volume changes. The topic is timely and highly relevant for various research branches including glaciology, hydrology, and climatology. I am very much in favor of seeing this manuscript published, and would like to make the following suggestions.

We really thank the positive comments provided for the reviewer to our work. We hope the new version of the paper will match the Journal requirements.

**Methods**

- The authors used state of- the art remote sensing and photogrammetric techniques to generate digital elevation models to estimate volume changes. The authors also applied state-of-the-art post-processing techniques (including co-registration, gap filling, outliers filtering, etc.) to provide a complete series of glacier elevation, volume, and area changes for the whole massif-volcano. However no information about the glacier area estimation.

Aerial photographs allow extracting surface area for each geodetic survey. These data were used to compute glacier-wide mass balances according to Equation 5. Ice-cap outlines and glacier boundaries were manually digitalized in stereo mode following the limits of the glacierized catchments. Thus, we estimate a total reduction of 42% in surface area for the whole ice cap, but also the surface area change for each period. These results were reported in Section 4.1, however, we agree with the reviewer and we include a specific figure of the surface area changes of each glacier in the supplementary material. Now, you can find in L192: *"Surface areas for each geodetic survey were manually digitalized in stereo mode following the boundaries of the glacierized catchments."*

- They also evaluate the effect of the morpho-topographic and climatic variables on glacier volume changes. However, in some sections, they mixed morpho-topographic-climate or vice-versa. In the title the use morpho-climatic. I suggest being consistent with the terms and clearly stating the variables evaluated.

**Volume to mass changes conversion:**

- The authors used one conversion factor (density) of ice volume change (850 kg m-3). However, very little discussion is associated with the choice of this number. Why just this value? Are the uncertainty ranges sufficiently? (±60 kg m-3). The authors also report that during the period 1965-1978 all the glaciers gain mass (moderate). Maybe it is possible to present density scenarios (e.g. Seehaus et al., 2019). For instance, a second scenario of two different conversion factors for areas below and above the ELA (e.g. Kääb et al. 2012).

Unfortunately, it is not possible to state density scenarios based on ELA because we do not know the ELA value at that time, however, we agree with the comment of the reviewer. In the new version of the manuscript we include two density assumptions (see Uncertainty analysis section). We also discuss in detail the implication of this assumption (see Results and discussion section).

You can now read: *"Second, regarding the uncertainty related to the density assumption, we analyze two extreme scenarios: First, we consider an average density recommended by Huss (2013) of $\bar{\rho}$ = 850 kg m$^{-3}$ with a plausible uncertainty range of $\sigma_\rho$ = ±60 kg m$^{-3}$. This value is appropriate for a wide range of conditions and when no information on firn pack changes is available (Huss, 2013; Zemp et al., 2013). However, moderate mass gains occurred in the second study period for which the conventional density assumption may not be true. Taking advantage of firn compaction data in two shallow core (mean depth ~14m) extracted from the summit of the Antisana volcano in February 1996 and November 1999, respectively (Calero et al., 2022; Williams et al., 2002), we propose a second scenario with an average density value of $\bar{\rho}$ = 564 ± 64 kg m$^{-3}$, indicating that the mass gain or loss was mostly comprising firn."*

We also include a section to discuss the sensitivity of the mass balance to the density assumption. Now you can read as follow:

*"4.2    Sensitivity of the geodetic mass balance to the density assumption*

*In most of the geodetic studies, when there is no information available about changes in firn pack it is strongly recommended to use a conservative density value such as the one proposed by Huss (2013), especially in periods of mass loss. However, in our glaciers, the second period (1965-1978) is characterized by mass gain and a density value close to the density of ice could led to an overestimation of the mass balance. Assuming a density of 850 kg m$^{-3}$ both in the accumulation and ablation areas for 1965-1978 period, the mass balance increase to 0.06 m w.e yr$^{-1}$ which is within the uncertainty of the mass balance. In addition, during 1998-2009 period, seven glaciers in the Antisana ice cap are close to equilibrium with a slightly positive or negative mass balance no matter what density scenario is assumed. Given the small difference between both assumptions, we decide to apply an average density value of 850 kg m$^{-3}$ when mass losses prevails, and when positive conditions are present we use an average density of 564 kg m$^{-3}$ according observational data in the summit of the Antisana ice cap (Calero et al., 2022; Williams et al., 2002)."*

**Uncertainties:**

- Overall, no details are mentioned about glacier area estimation or source. How did you obtain the glacier areas? How was the uncertainty of glacier mapping considered? No details about the uncertainty of the glacier area are included (not included in your error propagation equation).

We reply this in the first comment. We also include an uncertainty value related to the surface area glacier determination, now you can read in the Uncertainty analysis section: *"Fourth, uncertainty in surface area determination of glaciers $\sigma_{Sg}$ considers a buffer zone of 1-pixel surrounding the glacier boundary (**¡Error! No se encuentra el origen de la referencia.**), and is computed by following the same approach as used to determine the uncertainty when no elevation measurements are available (see Equation 8) but replacing $S_{g.void}$ for $S_{g.b}$ which is the surface area of the buffer zone. "*

**Specific comments:**

Title: I am not fully convinced with your title. I would suggest restructuring the title since this study signifies the first long-term geodetic mass balance /volume change, and also because Antisana ice cap more than "iconic" is a benchmark glacier for the inner tropics.

In the sake of clarity the title is changed to *"New insights into the decadal variability in glacier volume of a tropical ice-cap explained by the morpho-topographic and climatic context, Antisana, (0°29' S, 78°09' W)"*

Abstract: Please provide numbers of volume or mass changes for this section. Strong and slight mass loss can sometimes be subjective.

Ok, numbers were included

21 -> what about the climatic variables

The description is given in L25

80 -> it seems that it was an important eruption in 1800. 370 -> is there any signal of geothermal activity in the Antisana glacier? This could explains the surge event?, at least it is a factor that is should be considered since is an active volcano (although its last eruption was in 1800). Is there any fumarolic activity?

At our knowledge there is was not volcanic activity reported over the past 400 years (personal communication from ML Hall, 2014). The volcano has been considered as a dormant volcano for more than a century and there is no evidence for geothermal activity or a local decrease in ice due to hot streams on the glaciers and the surrounding terrains (personal communication from P Ramon, 2014). Nevertheless, we cannot reject with 100% of certainty that surge event is not related to an increase in basal melt due to heat transfer from the volcano. Unfortunately, heat fluxes have not been measured to confirm this hypothesis. If geothermal contribution exist, this would be very local.

Following your comment and a pretty similar one by Reviewer 1, we added this hypothesis in the manuscript:

- In section 4.3: *"In the present case, it could be hypothesized that sub-surface heating enhancing basal melt might be part of the triggers of this surge event, but no volcanic activity has been evidenced."*
- In the Conclusion: *"To our knowledge, no similar event has been reported in the tropics to date, thus more research is needed before being able to conclude on the internal (ice-flow dynamics) or external factors (climate, sub-surface heating due to volcanic activity) that triggered such an event."*

115 -> did you scan the negatives? or how was the digitalization process for the aerial photographs?

Yes, we have all the films scanned, also we find by the chance the original calibration reports for the cameras. What is normally very difficult to access. We clarify this point, and you can now read *"The aerial films were scanned at 14 µm resolution using an Intergraph PhotoScan TD system. All the calibration reports of each sensor were available, this information is essential to reconstruct the geometry of the sensor at the moment of the aerial acquisition."*

135 -> did you apply any correction (GCP points) to the Pleiades image? Some of the images present some displacement.

As is described in the manuscript, Pleiades imagery is oriented by using rational polynomial coefficients (RPCs) which were provided by the ancillary information of the satellite. A bias evaluation shows that Pleiades elevation data was displaced around 8 m above the ground relative to the 2009-Dem (i.e., geodetic reference), this is probably explained by the fact that the triangulation and geometric adjustment of Pleiades images was based on the RPC model without including GCPs. Thus, the Pleiades DEM had to be adjusted horizontally and vertically by performing a co-registration procedure proposed by Nuth and Kääb, (2011).

212 - 216 -> Please check this, you have included the internal ablation due to the heat transfer in the subglacial interface layer and due to heat released due to glacier dynamics in your uncertainties. However, I think this is not necessary. To my knowledge, the geodetic mass balance is providing the total glacier mass balance including internal ablation (Cogley et al., 2011). Hence the uncertainty from the geodetic estimation should be enough.

We agree with reviewer, the geodetic mass balance covers the internal and basal components of the surface balances. We are sorry for this mistake, we remove this part in the new version of the manuscript following your suggestions. We also update the uncertainty analysis section.

214 -> include the area error/uncertainty into your propagation equation. No details about the uncertainty of the glacier area mapping are included.

Please refer to our reply in the Uncertainties comments. We also update the uncertainty analysis section.

280 -> what do you consider as morpho-topographic features just an elevation profile? Please provide clear detail about the morpho -topographic -climate variables. In the title of your study, you just included the morpho-climatic. Please be consistent throughout the text.

The text was edited according your suggestion.

320 -> Table 4 -> the periods should be 1956-1965; 1965-1979, 1979-….etc…did you calculate the dhdt using these dates? You stated in line 237 that the time was not adjusted. I think that the results from â…€period and 1956-2016 should be included in your uncertainty estimation as well (e.g. Brun et al., 2017; Menounos et al., 2019 -systematic errors-).

Is well known that the best period for carrying out aerial surveys in this part of the Andes is during the less rainy season, when glaciers are free of snow cover (i.e., September to January) (see Figure 1). Four geodetic survey (1956, 1965, 1979 and 2016) were carried out very close to the beginning of the hydrological year (i.e. from December to January), therefore, we assume a fix-date reference for these years. For the other survey dates conducted at floatingdate reference (i.e., August 1997 and September 2009), the survey difference covers a time span of about four months. Only one small glacier G15α (0.28 km²) has monthly mass balance observations dating from the mid-1990s (Basantes-Serrano et al., 2016). Although the glaciers are very close to each other, there is not possible to assume a linear mass balance evolution based on the mass balance rates of G15α due to the variability of mass balance from glacier to glacier.

Additionally, the sum of the glacier mass balance calculated from the five sub-periods ($\Sigma_{sub-period}$) does not correspond exactly with the mass balances calculated for the full period (1956-2016). Note that the full period covers from Feb-1956 to Dec-2016, thus we assumed a fix-date reference of 61 years. Unlike Brun et al., (2017) and Menounos et al., (2019), however, in our study each period begins where the previous one ends, therefore the discrepancy between the ($\Sigma_{sub-period}$) and the geodetic balance of the full period cannot be explained by differences in survey dates but mainly by data gaps.

To evaluate the systematic error due to the mass-balance processes occurred between the date of the geodetic survey and the end of the hydrological year, we consider a linear glacier surface evolution hypothesis based on the geodetic mass balance. Now, you can read: *"[…] therefore no time adjustment is is possible and we kept the original dates for the mass balance estimations, this is called floating-date reference. To evaluate the systematic error due to the survey difference $\sigma_{t.ref}$, we assume constant monthly mass balance rates at the glacier surface based on the geodetic mass balance. Then, the monthly mass balance is multiplying by the number of months to match the hydrological year. The uncertainty due to the time reference is evaluated as the residual between the annual mass balance at floating-date and the annual mass balance at fixed-date.*

371 -> It is a confusing sentence. Ice flow dynamics are also a response of climate variations.

This sentence was removed.

405 -> table 5 -> Just morpho-topographic?  Please indicate what is Bm and 'Bm

Solar radiation was included in the caption. Now the declaration of the variables is in the caption of the table. Ok, the text is added to explain those variables *"Considering the mass balance, (Bm) is for all the glaciers in the ice cap and ('Bm) is excluding the outlier glaciers (G1a, G5 and G16)."*

I missed a comparison of your results with those from Braun et al., (2019) and Dussaillant et al., (2019). Although they used RGI_V6 glacier outlines to estimate volume change over a limited period, it is a good opportunity to check their number with more high-resolution data as you have shown here.

We evaluate the dh/dt coverage computed from ASTERIX technique by Dussaillant et al., (2019) and from this study to make a comparison for similar periods. This information was added in supplementary materials as Appendix S1. It is worth mentioning that this is an issue that is currently evaluated in hydrological terms for the entire ice cap and it will be presented in a future work. You can read now:

*"**Appendix S1.** Comparison with previous estimates of elevation change*
*To evaluate the agreement between the elevation changes observed in this study and previous geodetic estimates from Dussaillant et al., (2019), we select a portion of 9 km2, in the western*

*side of the ice cap. See the Randolph Glacier Inventory (RGI) v6.0 for more details. This location was selected because of the limited number of data voids in both datasets (Table S2).*

**Table S2.** *Average elevation change and percentage of surface area covered by dh-samples.*

| Dussaillant et al., (2019) | Data coverage (%) | This study | Data coverage (%) |
|---|---|---|---|
| -0.95 m a⁻¹ (2000-2009) | 75 | -0.28±0.06 m a⁻¹ (1998-2009) | 85 |
| -0.07 m a⁻¹ (2009-2018) | 65 | -0.17 m a⁻¹±0.06 (2009-2016) | 84 |
| -0.28 m a⁻¹ (2000-2018) | 99 | -0.25 m a⁻¹±0.06 (1998-2016) | 99 |

*For the full period (1998-2018) we found a good agreement in the elevation change rates, which is not the case for the sub-periods (1998-2009) and (2009-2018) where a noticeable discrepancy is observed (Fig S6). Unlike 1998-2018 period, the dh/ht maps obtained by ASTER imagery tend to be noisy during short periods when the data gaps are around 30%. Data gaps may be related to the presence of cloud or snow cover that prevent the determination of reliable elevation changes. These issues have been previously reported over large Patagonian ice fields (Dussaillant et al., 2019), and could also be the case for a region as humid as the inner tropics where Antisana ice cap is located. However, in spite of the difference observed in ASTERIX data, they are able to capture the similar trend of elevation change rate obtained by this study.*

*Negative conditions observed in Dussaillant et al., (2019) for the 2000-2009 period are suspicious and does not match with the conditions observed on Antisana glaciers in a similar period (1998-2009), this period has a 25% of data voids resulting in an underestimation of the mass losses."*

**Figures:**

The figures are clearly meant to support the overall study, but they also present some issues for the reader. Figures 1 and 2 -> maybe both images can be merged, and the insert table can be inserted as a normal table.

Figures 1 and 2 present a lot of information. The first one is about the local context the second one is about the regional context. We believe that merging the two images can lead to a misinterpretation and to keep the clarity of the data we prefer to keep the images separated.

Figure 3 -> It is difficult to follow the colors. Is it possible to change the brightness of the plot? it is not possible to identify the colors (opaque). In the period 1956-2016, there are data gaps mainly in glaciers from G4 to G7. I was wondering how you managed the samples in these accumulation areas (gray). The same for 2010-2106 period.

We agree with the comment. In the new manuscript we update the color scale by increasing the colors to be able to distinguish the ranges of elevation change. The data gaps in the elevation difference layers are present in the upper reaches of the glaciers (accumulation zone), in areas with: i) low contrast (e.g., snow cover patches), ii) image saturation, iii) cloud coverage, and rugged topography (slope > 45°). Thus, we removed dh outliers considering a similar approach applied by (e.g., Braun et al., 2019; Brun et al., 2017).

We also evaluate the systematic bias due to data gaps, and we confirm that all the dh-samples are randomly distributed over the glacier. A new figure has been added in supplementary materials (Fig. S2). In the sake of clarity, you can now read in the Uncertainty Analysis section (point fourth): *"In addition, it is worth mentioning that the dh coverage for all periods are evenly distributed over the glacier surface, which reduces the likelihood of inducing some spatial biases in the quantification of glacier elevation changes (Fig. S2 in supplementary materials)."*

**References**

Braun, M. H., Malz, P., Sommer, C., Farías-Barahona, D., Sauter, T., Casassa, G., Soruco, A., Skvarca, P. and Seehaus, T. C. (2019). Constraining glacier elevation and mass changes in South America, Nat. Clim. Chang., doi:10.1038/s41558-018-0375-7.

Brun, F., Berthier, E., Wagnon, P., Kääb, A. and Treichler, D. (2017). A spatially resolved estimate of High Mountain Asia glacier mass balances from 2000 to 2016, Nat. Geosci., 10(9), 668–673, doi:10.1038/ngeo2999.

Dussaillant, I., Berthier, E., Brun, F., Masiokas, M., Hugonnet, R., Favier, V., Rabatel, A., Pitte, P. and Ruiz, L. (2019).Two decades of glacier mass loss along the Andes, Nat. Geosci., doi:10.1038/s41561-019-0432-5, 2019.

Menounos, B., Hugonnet, R., Shean, D.,Gardner, A., Howat, I., Berthier, E., et al. (2019). Heterogeneous changes in western North American glaciers linked to decadal variability in zonal wind strength.Geophysical Research Letters,46, 200–209. https://doi.org/10.1029/2018GL080942

Seehaus, T., Malz, P., Sommer, C., Lippl, S., Cochachin, A., and Braun, M. (2019). Changes of the tropical glaciers throughout Peru between 2000 and 2016 – mass balance and area fluctuations, The Cryosphere, 13, 2537–2556, https://doi.org/10.5194/tc-13-2537-2019.